TOOLS

# High-precision targeting workflow for volume electron microscopy

Paolo Ronchi[1], Giulia Mizzon[1], Pedro Machado[1], Edoardo D'Imprima[2], Benedikt T. Best[3], Lucia Cassella[4,5], Sebastian Schnorrenberg[6], Marta G. Montero[7], Martin Jechlinger[7], Anne Ephrussi[4], Maria Leptin[3], Julia Mahamid[2], and Yannick Schwab[1,7]

**Cells are 3D objects. Therefore, volume EM (vEM) is often crucial for correct interpretation of ultrastructural data. Today, scanning EM (SEM) methods such as focused ion beam (FIB)–SEM are frequently used for vEM analyses. While they allow automated data acquisition, precise targeting of volumes of interest within a large sample remains challenging. Here, we provide a workflow to target FIB-SEM acquisition of fluorescently labeled cells or subcellular structures with micrometer precision. The strategy relies on fluorescence preservation during sample preparation and targeted trimming guided by confocal maps of the fluorescence signal in the resin block. Laser branding is used to create landmarks on the block surface to position the FIB-SEM acquisition. Using this method, we acquired volumes of specific single cells within large tissues such as 3D cultures of mouse mammary gland organoids, tracheal terminal cells in *Drosophila melanogaster* larvae, and ovarian follicular cells in adult *Drosophila*, discovering ultrastructural details that could not be appreciated before.**

## Introduction

The blooming of new technologies over the last decade has dramatically increased the value of EM for cell biology. Volume scanning EMs (SEMs) have opened the possibility of visualizing large volumes (Peddie and Collinson, 2014; Titze and Genoud, 2016). Serial block-face SEM (SBEM; Denk and Horstmann, 2004) and focused ion beam (FIB)–SEM (Heymann et al., 2006, 2009; Knott et al., 2008; Hekking et al., 2009) offer the possibility of imaging volumes at nanometer resolution in a semi-automated way. Both instruments combine iterative slicing and SEM imaging. While SBEM uses a diamond knife to remove thin sections from the block surface, in the FIB-SEM an ion beam (most often Gallium ions) is used to ablate thin layers of material. In both cases, the electron beam is scanned onto the freshly exposed sample surface to produce images. The iteration of imaging and milling over thousands of cycles generates a stack of images that are then digitally combined to reconstruct a volume (Narayan and Subramaniam, 2015). For both techniques, the achievable X,Y resolution is comparable (3–4 nm). However, the use of an ion beam enables finer sectioning resolution, down to 3–4 nm (Wei et al., 2012; Xu et al., 2017; Hoffman et al., 2020; Müller et al., 2021). For this reason, while SBEM is mostly suitable for the imaging of very large volumes (tissues or small organisms) in a nonisotropic fashion, FIB-SEM is currently

considered best suited to automatically acquire relatively small volumes (ranging from subcellular to a few cells) at high isotropic resolution.

Specific sample preparation protocols have been designed for FIB-SEM and SBEM (e.g., Deerinck T.J., et al. 2010. *Microsc. Microanal.* https://doi.org/10.1017/S1431927610055170; Maco et al., 2013; Hua et al., 2015). Common to all of them is the requirement to introduce high amounts of heavy metals en bloc during sample processing. The electron-scattering properties of heavy metals such as osmium, uranium, and lead produce high image contrast. At the same time, metals improve sample conductivity, which is crucial during electron imaging. Another important aspect of sample preparation concerns the choice of embedding media. The best milling performances with the FIB-SEM have been achieved so far with hard and rigid epoxy resins such as Epon812, Hard Plus, Spurr's, and Durcupan (Kizilyaprak et al., 2015).

When approaching a large specimen for volume EM (vEM), restricting the acquisition to a subvolume of interest is often necessary. To image a defined structure (e.g., a specific cell in a tissue), precise targeting allows the optimization of the acquisition time and the amount of data generated, therefore increasing the throughput of such experiments. To identify the

[1]Electron Microscopy Core Facility, European Molecular Biology Laboratory, Heidelberg, Germany; [2]Structural and Computational Biology Unit, European Molecular Biology Laboratory, Heidelberg, Germany; [3]Directors' Research, European Molecular Biology Laboratory, Heidelberg, Germany; [4]Developmental Biology Unit, European Molecular Biology Laboratory, Heidelberg, Germany; [5]Faculty of Biosciences, Heidelberg University, Heidelberg, Germany.; [6]Advanced Light Microscopy Facility, European Molecular Biology Laboratory, Heidelberg, Germany; [7]Cell Biology and Biophysics Unit, European Molecular Biology Laboratory, Heidelberg, Germany.

Correspondence to Paolo Ronchi: ronchi@embl.de; Yannick Schwab: schwab@embl.de; P. Machado's present address is Centre for Ultrastructural Imaging, King's College London, London, UK; B.T. Best's present address is Centre for Organismal Studies, Heidelberg University, Heidelberg, Germany; M. Jechlinger's present address is MOLIT Institut gGmbH, Heilbronn, Germany.



region of interest (ROI), either morphological cues or fluorescence can be used, depending on the application. When a structure or cell of interest can be labeled by fluorescent dyes or proteins, correlative light and EM (CLEM) is the most popular choice to guide EM acquisition or to integrate the ultrastructural information offered by EM with the molecular identity labeled by fluorescence (Mironov and Beznoussenko, 2009; Caplan et al., 2011; de Boer et al., 2015; Bykov et al., 2016). Although many CLEM workflows are available, using fluorescent signals to target specific objects within a resin-embedded specimen is still a challenging task. Indeed, both the high concentrations of heavy metals and the epoxy resins required for volume SEM sample preparation affect the fluorescence and result in non-optimal properties for light microscopy (LM). Metals quench fluorophores in their vicinity, while heat-polymerized hydrophobic epoxy resins dehydrate and denature fluorescent proteins (FPs; Paez-Segala et al., 2015). Therefore, pre-embedding LM is often necessary for FIB-SEM targeting. However, staining, dehydration, and embedding induce anisotropic distortions to the sample (Zhang et al., 2017). As a consequence, the location of a structure to be acquired by vEM cannot be predicted with sufficient precision from pre-embedding LM. A third imaging modality (e.g., x-ray microscopic computed tomography) becomes necessary to reveal the local distortion introduced during sample preparation (Karreman et al., 2016). This has proven helpful to predict the position of ROIs after sample preparation. However, this workflow is complicated and requires access to additional expensive equipment.

To overcome the problem of sample deformation and to precisely assign the location of a fluorescently tagged structure in an EM image, strategies were developed to preserve fluorescence during sample preparation and allow postembedding CLEM (Nixon et al., 2009; Kukulski et al., 2011; Peddie et al., 2014; Biel et al., 2003). In such approaches, fluorescence preservation is enabled by reducing the amount of heavy metals in the sample. These protocols avoid osmium and use small amounts of uranyl acetate (UA) to stain the sample. In addition, the use of methacrylate resins (e.g., Lowicryl HM20; Armbruster et al., 1982), which are less hydrophobic than epoxy resins and can be UV-polymerized at low temperatures, circumvents the heat-induced denaturation of the FPs. These approaches have been developed and used mostly for transmission EM (TEM; Nixon et al., 2009). The possibility of imaging the same field of view (FOV) on the same sections at the light microscope before moving to the TEM increases the accuracy of correlation for on-section CLEM (Kukulski et al., 2011; Avinoam et al., 2015). However, TEM imaging techniques have limited power for acquiring large volumes in 3D as they depend on tedious serial sectioning and large-scale imaging of serial sections (Mathew et al., 2020). Therefore, workflows that combine postembedding LM and an automated volume SEM are required to increase the throughput of volume CLEM experiments.

Recent publications showed good results in FIB-SEM acquisition of samples that were high-pressure frozen, freeze substituted (FS) with low amounts of UA, and embedded in acrylic resins (Höhn et al., 2015; Porrati et al., 2019). Based on these observations, we have established an easy and robust workflow that allows targeted FIB-SEM imaging of fluorescently labeled structures in a large volume (∼3.14 mm² × 0.2/0.4 mm depth). We show that FIB-SEM targeting can be achieved with micrometer precision based exclusively on fluorescence, without relying on anatomical or morphological features. Despite the low amount of heavy metals in the sample and the lower hardness of the resin used, the imaging quality enabled fine ultrastructural analysis, at par with traditional protocols (Polilov et al., 2021), demonstrating that FIB-SEM is compatible with a large spectrum of sample preparation procedures.

## Results

### Sample preparation

To develop a sample preparation strategy that would best combine EM ultrastructure quality and fluorescence preservation for targeting, we adapted the protocols previously optimized for on-section CLEM experiments (Hampoelz et al., 2016, 2019; Wong et al., 2020; Lee et al., 2020). We high-pressure froze the samples (cultured cells in suspension, mammary gland organoids in Matrigel, *Drosophila* ovaries, and dissected *Drosophila* larvae) and FS them with 0.1% UA in dry acetone (Table S1). In contrast to other reports (Peddie et al., 2014), the addition of water was not necessary to preserve fluorescence, even though we cannot rule out that water contamination, via condensation, could have been introduced together with the cold high-pressure frozen planchettes in the FS cocktail. After 72 h incubation at –90°C, the temperature was increased to allow the UA to stain the biological material. We found that an optimal concentration of UA in the sample (the best compromise between EM contrast and fluorescence preservation) was achieved by increasing the temperature to –45°C at a speed of 3°C/h and then incubating the samples in the UA solution for an additional 5 h at –45°C. Compared with the original on-section CLEM protocols (e.g., Kukulski et al., 2011), the temperature rise rate after the FS –90°C step was slower (3°C/h vs. 5°C/h). This was crucial in our hands to achieve satisfactory contrast with the samples we used. For instance, in *Drosophila* ovaries, membranes appeared with negative contrast with a rate of 5°C/h (not shown). The samples were then rinsed with pure acetone before infiltration with the resin Lowicryl HM20. This sample preparation method preserved the fluorescence of the samples, especially for red FPs, including mCherry and DsRed. We could image fluorescence signals at a depth of several hundreds of microns within the resin block when scanning with a confocal microscope over the entire block (Fig. 1, A, E, K, and O). Moreover, this sample preparation was compatible with FIB-SEM acquisition. We could achieve good imaging and milling quality for large volumes (up to ∼80 μm × 60 μm × 80 μm; Fig. 1, B, F, L, and P), with sufficient contrast to visualize subcellular structures, when imaging at 8- or 10-nm voxel size. For example, we were able to visualize not only membrane-bound organelles such as mitochondria (Fig. 1, C, I, and T; cristae visible in Fig. 1, C and I), the Golgi apparatus (Fig. 1, G and M), multivesicular bodies (MVBs; Fig. 1, H and S), and the ER (Fig. 1 R) but also membrane invaginations (Fig. 1 Q), nuclear pores (Fig. 1 N),

Figure 1. **Sample preparation provides optimal fluorescence preservation and FIB-SEM imaging quality. (A–D)** HeLa cells expressing H2B-mEGFP (green) or H2B-mCherry (red). **(A)** Confocal image of the resin block. **(B)** FIB-SEM slice of the dividing cells shown in A, acquired at 10-nm isotropic voxel size. Note that the imaging plane at the FIB-SEM is orthogonal to the confocal one. **(C and D)** High-resolution details of FIB-SEM acquisitions. In C, a group of mitochondria with visible cristae; in D, a midbody with cytoskeleton bundles. **(E–J)** Primary mammary gland organoids expressing H2B-mCherry (red). **(E)** Confocal image acquired from the resin block. In red, the mCherry signal, overlaid to the bright-field image. **(F)** Slice of the FIB-SEM volume of the entire organoid shown in E, acquired at 15-nm isotropic voxel size. **(G–J)** High-magnification details of single-cell volumes acquired from other organoids at 8-nm

isotropic pixel size. In G, Golgi complex; in H, MVBs, with visible single vesicles in the lumen; in I, a mitochondrion (asterisk) and a bundle of cytoskeleton filaments (probably microtubules, arrowhead); in J, a centrosome with the two centrioles highlighted by arrowheads. **(K–N)** *Drosophila* trachea terminal cell expressing cytoplasmic DsRed. **(K)** Confocal slice acquired from the resin block. In green, autofluorescence of the tissue (including the tracheal tube). In red, DsRed, specifically expressed by trachea cells. The arrowhead indicates the cell shown in L. **(L)** Slice of the FIB-SEM volume of a portion of the fluorescent cell shown in K, acquired at 10-nm isotropic voxel size. **(M and N)** Details of the same volume, showing the Golgi apparatus and mitochondria (M) and nuclear pores in top view, at the nuclear envelope (N). **(O–T)** *Drosophila* ovarian FCs, with clonal expression of *Dhc* RNAi and CD8-mCherry. **(O)** Confocal image acquired from the resin block. In red, the CD8-mCherry signal, overlaid to the bright-field image. Oocyte and FCs are indicated. **(P)** Slice of the FIB-SEM volume of the same cell shown in O, acquired at 10-nm isotropic voxel size. **(Q–T)** Details of the same volume: in Q, invaginations of the oocyte plasma membrane; in R, area rich in ER cisternae in an FC; in S, MVBs; and in T, mitochondria.

centrioles (Fig. 1 J), microtubule bundles in the midbody (Fig. 1 D), and single microtubules (Fig. 1 E).

### Fluorescence imaging in block

As fluorescence imaging in blocks has not been well characterized so far, we set out to quantify the behavior of monomeric EGFP (mEGFP) and mCherry fluorescence in the block using a laser scanning confocal microscope. For this analysis, we used blocks prepared as described above containing a mixed suspension of HeLa cells expressing either histone 2B (H2B)–mEGFP or H2B-mCherry. First, we compared the fluorescence levels of the embedded sample with the same cells after formaldehyde fixation. We found that we needed ~3× more laser intensity for mCherry and ~20× more for mEGFP to image embedded compared with nonembedded samples. Next, as hydration was shown to influence fluorescence imaging of sections (Peddie et al., 2017), we tested the effect of water on the fluorescence intensity of H2B-mEGFP and H2B-mCherry–positive nuclei. To this aim, we placed the block in a glass-bottom dish (Mattek) and measured the fluorescence intensity of the two fluorophores at the surface before adding water and at different incubation time points. In the absence of water, mEGFP signal could not be detected above the background, whereas mCherry-positive nuclei were visible (Fig. 2, A and B). Water boosted both fluorescence signals, and this effect increased over time (Fig. 2, A and B). We further characterized the fluorescence intensities at different depths inside the block and found that both signals exhibited an intensity decay in imaging planes below the surface (Fig. 2, C and D). However, while we were able to detect H2B-mCherry fluorescence from the entire depth of the block, mEGFP detection was limited to a few micrometers from the surface, where nuclei were exposed to water (Fig. 2, C and D). We further noticed that the signal was resistant to photobleaching during imaging (Fig. 2 C), in agreement with previous experiments on sections (Peddie et al., 2014). To quantify the photobleaching of mEGFP and mCherry in block, we repeatedly scanned confocal sections ~2 μm from the block surface with the same laser settings used for imaging (Fig. 2, C and D) and found that after 250 sequential scans, the signal dropped by <40%. This behavior was similar for both fluorophores.

A quantitative fluorescence characterization performed on mouse primary mammary gland–derived organoids expressing H2B-mCherry gave similar results (Fig. S1). Qualitatively, we observed similar behavior for all the samples and red FPs used in this study. In summary, although the fluorescence signal decreases with the distance from the block surface, the resistance to photobleaching allows targeting of structures throughout the entire high-pressure frozen sample volume using relatively high photon doses.

### Targeting strategy and FIB-SEM imaging

To set up our 3D targeting strategy, we used the 3D culture of mouse primary mammary gland organoids expressing H2B-mCherry. From a technical viewpoint, imaging organoids is particularly interesting because it requires a multiscale approach to resolve their overall architecture (spheroids up to 100 μm in diameter), as well as the structure of single cells at high resolution. Moreover, the targeting for FIB-SEM acquisition in a 3D culture is particularly challenging because of the lack of features visible in the SEM before the organoid is exposed.

After preparing the samples by high-pressure freezing and FS as described above, we separated the resin blocks from the aluminum planchettes and polished the block surface by removing a few microns with a diamond trimming knife (Cryotrim 90; Diatome). This removes the rings imprinted from the planchette, which could diffract the laser light, creating artifacts during the subsequent confocal acquisition. Next, we manually trimmed the block with a razor blade, giving it an asymmetric shape, which facilitates orientation in later steps. The block face was kept perpendicular to the long axis of the block and large enough for it to stand upright on the front face during imaging. Next, the block was placed in a glass-bottom dish, with the sample side on a drop of water. With this setup, we were able to acquire a tiled Z-stack confocal scan to cover the entire part of the block containing the biological sample (typically ~1,500 μm × 1,500 μm × 200/400 μm) and could identify the fluorescent targets (Fig. 3; Fig. 4, A and I; Fig. S2; Fig. S3, A and B; Fig. S4, A and B; Video 1; and Video 2). To increase the acquisition speed of such a large volume, we used a relatively low XY resolution (~500-nm pixel size), a large confocal optical slice (2–3 Airy Units), and 10-μm Z increment for the first scan.

FIB-SEM imaging requires precise Z targeting because ion milling quality degrades when moving deeper from the surface of the block. Therefore, it is beneficial to bring the target as close as possible to the block surface. Our targeting approach relies on cycles of confocal scans of the block and measurements of the distance of the target from the surface, followed by removal of the resin above the target with a trimming diamond knife (Fig. 3 and Fig. 4).

After identification of the target, the first trimming step was performed. In view of the low Z resolution of the fluorescence acquisition and inaccuracies due to mismatches in refractive indices of the materials, we did not remove the entire predicted

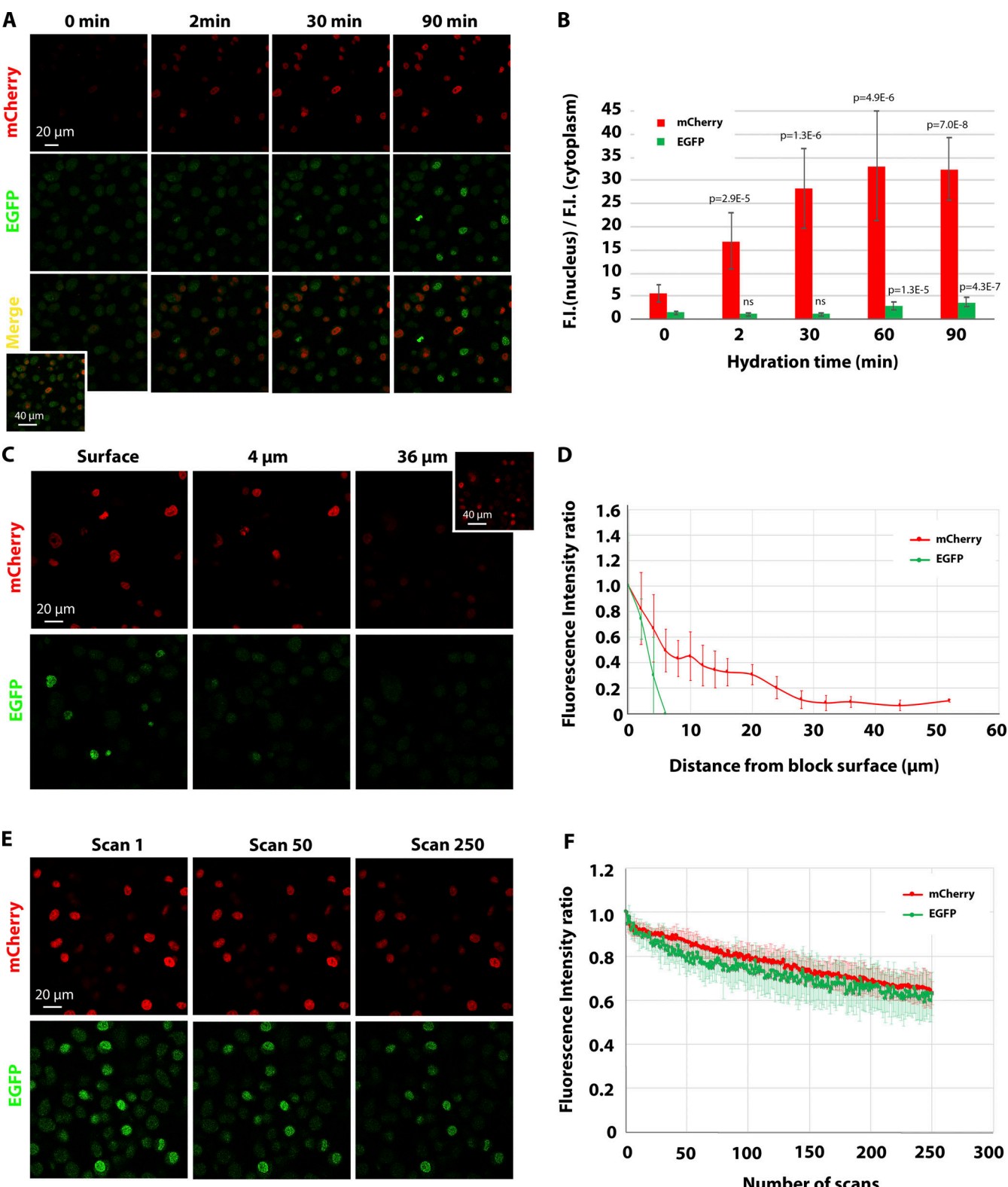

Figure 2. **Characterization of the behavior of mEGFP and mCherry fluorescence in resin block. (A)** Confocal imaging of H2B-mCherry and H2B-mEGFP in the resin block upon different time of incubation of the block in water. The same confocal volume was acquired for each condition with the same settings. Note that at 0 min, mCherry fluorescence is low but detectable, whereas mEGFP cannot be detected above the background (see inset in the merge, where the intensity has been digitally amplified). **(B)** Histogram showing the ratio between the fluorescence intensity (F.I.) in the nucleus and the fluorescence intensity in the cytoplasm (background) for H2B-mCherry and H2B-mEGFP. The average value of 10–12 cells for mCherry and 5–15 for mEGFP ± SD is shown. t test was used to compare the ratio at each time point with the one at 0 min. P value is shown. **(C and D)** Fluorescence intensity measurements of H2B-mCherry and H2B-mEGFP in a confocal stack. A representative example is shown in C. Note that while mEGFP fluorescence drops to the background level past the water-

exposed nuclei, the mCherry signal remains visible at higher depths (see inset, where the signal has been digitally amplified). The integrated fluorescence intensity of 9–12 nuclei per confocal slice was measured for each channel, and the average (±SD) is plotted in D in relation to the distance from the block surface. The data are normalized to the average fluorescence intensity of the nuclei at the surface. **(E and F)** Photobleaching behavior of H2B-mCherry and H2B-mEGFP in a resin block. An FOV containing both cell lines was consecutively scanned 250 times with the same settings used for standard imaging. Images are shown in E, and quantification of the integrated fluorescence intensity of 10 nuclei from three independent FOVs for each channel (±SD) is shown in F.

excess resin thickness, but left a considerable buffer (typically, 30–40 µm). Subsequent imaging with progressively better Z resolution and trimming cycles allowed us to get increasingly closer to the target. We normally executed this approach in three steps, which gradually brought the target within 30–40, then 10–15, and finally 2–5 µm from the surface (Fig. 3 and Fig. 4 C). Bringing the fluorescence target closer to the surface improved the imaging quality, allowing us to identify small targets with better precision (e.g., single mitotic cells in an organoid; Fig. 4, I–N).

After the final trimming step, the target structure is located at an optimal depth. However, the block surface is millimeters in size, while XY targeting for FIB-SEM needs to be accurate on the micrometer scale. Indeed, milling and imaging of large volumes with a FIB-SEM is very time-consuming, and long acquisitions often result in imaging instability. Therefore, a trench as small and precise as possible in XY is desirable. In the absence of landmarks at the surface of the trimmed block, as was the case for the organoid culture, positioning the target volume in X and Y can only be done using the distant block edges as a reference. Alternatively, landmarks can be manually introduced, for instance, by scratching marks on the block surface. However, both approaches produce limited accuracy. To facilitate precise XY targeting, we implemented a universal workflow by branding the surface of the block by two-photon laser. To this aim, we tested several wavelengths, laser powers, dwell times, and repetition numbers. The best experimentally determined setup with our microscope is described in Materials and methods. Branding results in an embossed feature that can be easily identified at the SEM, and we used it as a landmark to position the FIB-SEM acquisition window (Fig. 3; and Fig. 4, D and E). Creating asymmetric shapes can be useful for orientation (see, for instance, Fig. S2 and Fig. S3).

The targeting of a volume of interest as described here, including imaging, finding the target, trimming, and branding, was typically achieved in 3–4 h.

FIB-SEM imaging of a Lowicryl-embedded specimen has rarely been described but showed in our hands satisfying results with standard imaging parameters (see Materials and methods for details) and special attention to keeping a stable FOV. Similar to the behavior of other resins used in FIB-SEM, Lowicryl was sensitive to electron beam exposure. Changes in the FOV dimension led to milling instabilities (i.e., curtaining artifacts), which were resolved only after a few sections. However, using the confocal volume to predict the exact location of the FOV to be acquired, we were able to define and keep the acquisition window stable throughout the acquisition, therefore solving the problem. The SEM images showed adequate contrast despite the minimal amount of heavy metals in the sample, and we were able to visualize membrane-bound organelles and other subcellular structures within single (or few) cells imaged at 8–10-

nm isotropic voxel size. A compromise in resolution had to be made for a larger FOV in order to reduce the acquisition time and avoid milling instabilities due to an excessive electron dose. For instance, acquisition of full organoids was possible at 15 nm × 15 nm × 15/20 nm voxel size.

This method proved to be suitable for a large spectrum of applications, ranging from large volumes (e.g., entire mouse primary mammary gland organoids, Fig. 4, A–H; and Video 1) to single cells (mitotic telophase within a whole 3D cell culture, Fig. 4, I–N; and Video 2; or follicle cells [FCs] in *Drosophila* ovary, Fig. S4).

Multiple structures, distributed throughout the surface of the block but located at the same depth, can be easily targeted with this approach. We also tested the possibility of imaging structures located at different depths. Recent work showed that fluorescence can be recovered in resin after SEM acquisition if sections are rehydrated (Peddie et al., 2017). We tested if we could target a second structure after the first one had been imaged (and milled) by FIB-SEM (Fig. S2). For this experiment, we used a block containing *Drosophila* ovaries. Simultaneous expression of CD8-mCherry and dynein heavy chain siRNA was induced in sparse FC clones in a mosaic fashion. After a confocal scan of the block, we identified two groups of fluorescent cells located at different depths from the surface (Fig. S2 C). After targeting and acquiring the cluster positioned closer to the surface (segmented in green in Fig. S2, C and D), we aimed for the second group. We thus removed the part of the block containing the first acquired volume using a razor blade (Fig. S2 G). This was necessary to avoid damaging of the trimming knife due to the hardening of the milled resin. Confocal imaging of the remaining part of the block confirmed that the second target was still visible and comparable in brightness to its level before FIB-SEM of the first target (Fig. S2 J). We therefore approached and imaged this second group of cells following our workflow. This experiment shows that several structures of interest, also located at different depths, can be sequentially targeted within a single block.

## Acquisition of *Drosophila* larval tracheal cells

Having developed and characterized the method, we applied it to investigate biological samples that are otherwise difficult to approach by vEM. The first problem we addressed was the characterization of terminal cells of the tracheal system in *Drosophila* larvae. These cells form branches at the surface of oxygen-demanding tissues, such as muscles. As epithelial cells, the basal membrane of terminal cells faces the target tissue and forms the outside of the branches, while their apical membrane is invaginated to form an intracellular tube. The cells are supported by a collagenous ECM on the basal side and a chitin-containing ECM on the apical side (aECM; Öztürk-Çolak et al.,

High pressure freezing   Freeze substitution

sample preparation

confocal

- target identification

- measure depth from surface

stepwise trimming

1. coarse localization
   (30-40 $\mu m$)
2. medium targeting
   (10-15 $\mu m$)
3. fine targeting
   (2-5 $\mu m$)

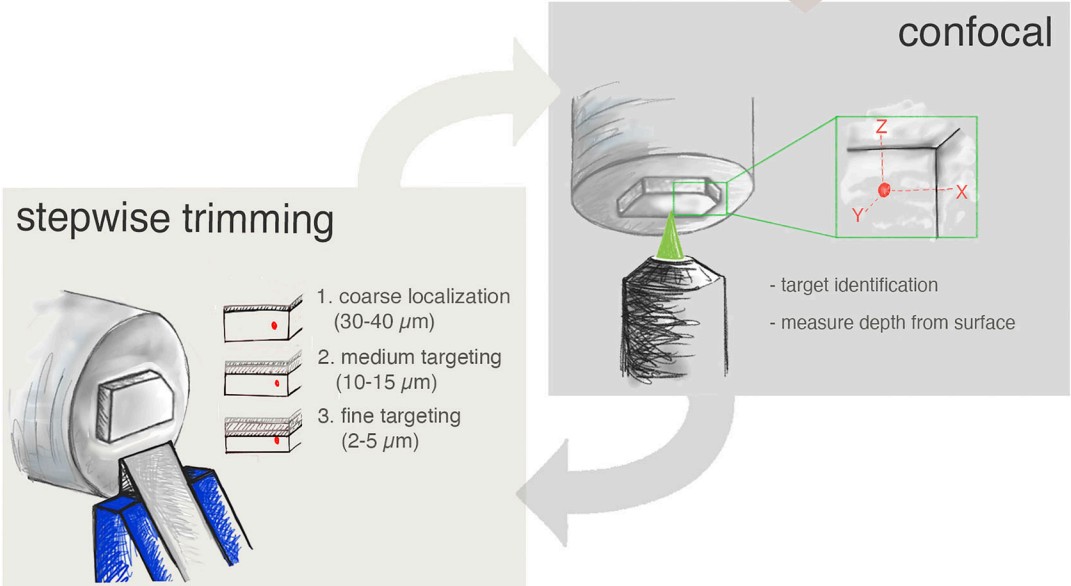

2pi branding

SEM view

trench milling

FIB

Laser branding
Platinum coating
Trench

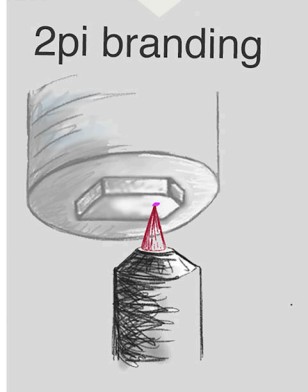
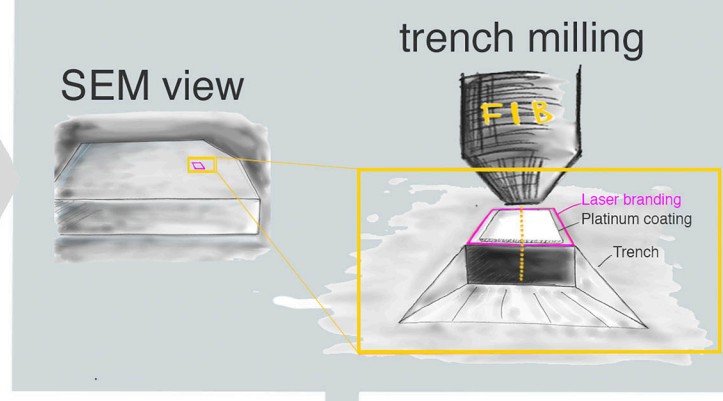

image acquisition &
volume registration

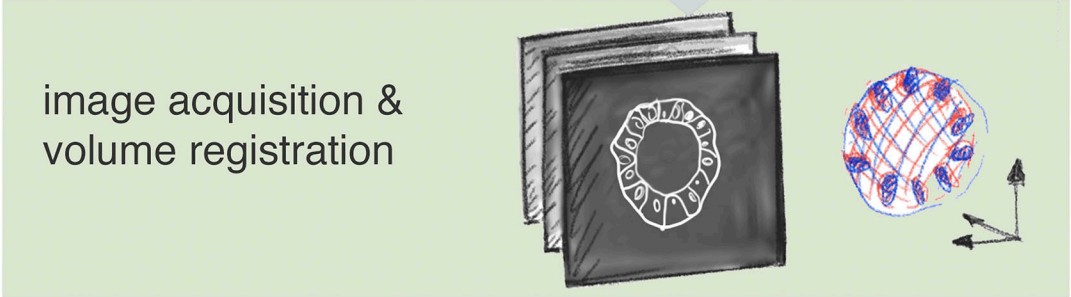

Figure 3.   **Schematic representation of the workflow.** From the top: Sample preparation consists of high-pressure freezing and FS. Second row: cycles of confocal acquisition and stepwise trimming at the ultramicrotome to progressively assess the depth of the target relative to the block surface. Normally, three

iterations were sufficient, reaching each time the approximate distance in Z from target as indicated in the figure. Third row: The block surface is marked by two-photon (2Pi) branding (magenta). Preparation for FIB-SEM consists of placing the platinum coating and trench milling with the ion beam in the vicinity of the branded mark. Finally, FIB-SEM imaging and image processing in the last row include registration of the volumes obtained by FIB-SEM and confocal acquisition.

2016a). The tracheal aECM forms ridges known as taenidia, which line the perimeter of the tubes and confer the physical rigidity that prevents the tubes from collapsing. Previous ultrastructural characterization of these cells was limited to 2D TEM at embryonic development or the earliest larval stages (Itakura et al., 2018; Öztürk-Çolak et al., 2016b; Jones et al., 2014; Nikolova and Metzstein, 2015). The ultrastructure of the tracheal ECM has thus never been observed at the latest larval stage, when the terminal cells undergo most of their growth (JayaNandanan et al., 2014).

Due to the size of the animal and the small number of terminal cells, their analysis by vEM requires a precise targeting strategy (Fig. S3, A and B). We used flies expressing DsRed in all tracheal cells using the Gal4/UAS transgenic expression system. The larvae were dissected and prepared as described in Materials and methods. The fluorescence was preserved in all cases (Fig. 1 and Fig. S3), and we identified several ROIs in each sample, where the nuclei of the cells and/or branching points of the tracheal tube were visible (Fig. S3 and Fig. 5). The volumes of interest ranged from a few microns from the surface to up to 80-µm depth within the block. Our targeting method was successfully used (Fig. S3) and allowed us to image eight cells with high precision by FIB-SEM (Fig. 5, A and B). The analysis of such areas revealed for the first time the aECM organization at the branching points (Fig. 5, A, C, and D) and a novel topology of the taenidial ridges (Fig. 5, C and D; and Video 3). While in all previously published studies (embryos and first-instar larvae) the taenidia appear as knobs (Öztürk-Çolak et al., 2016b; Itakura et al., 2018) or ridges (Nikolova and Metzstein, 2015) in their cross section, our data revealed that the shape in the third larval stage is reminiscent of teeth (Fig. 5, C, D, and E). The aECM of larger multicellular branches showed the same tooth-like structures (data not shown). This was consistent across all samples, suggesting that the morphology of the tracheal aECM of third-instar larvae differs from that of the earlier developmental stages. We observed that tracheal and muscle cells each have their own basal lamina and that the contact surface therefore shows a double layer of these basal ECM sheets separating the plasma membranes (Fig. 5 E and Video 3). This is in contrast to the organization in the wing disc, where tracheal branches are found encapsulated within the target tissue's basal lamina (Guha et al., 2009). Moreover, when we characterized the cells more closely, we observed in some cases hallmarks of cells undergoing active membrane trafficking on the basal side (Fig. 5 F). While it is known that tracheal cells receive proteins to secrete into the apical tube from other organs and thus must take them up on their basal membrane first (Dong et al., 2014), this was only shown during embryonic development. It is unknown what function basal membrane trafficking might serve at this late larval stage. Moreover, we often observed structures that might represent fusion events of carrier vesicles containing electron-scattering material with the apical plasma membrane (Fig. 5 G and Video 3), consistent with the delivery of material that constitutes the aECM of the tracheal tube.

## Acquisition of *Drosophila* ovarian FCs

The *Drosophila* follicular epithelium is a monolayer of somatic epithelial cells that encapsulates the developing germline cyst and is necessary to induce polarization and growth of the oocyte during oogenesis. During mid and late oogenesis, the FCs differentiate into a secretory epithelium with an apical domain proximal to the oocyte and are responsible for the apical secretion of a subset of yolk proteins and all three eggshell components (vitelline membrane, wax layer, chorion). Generating genetic mosaics provides the possibility to study the effect of deleterious mutations in FCs, which would otherwise disrupt embryonic development. The FLP-out technique allows the permanent expression of a gene of interest in a small subset of cells upon a short heat-shock treatment (Struhl and Basler, 1993; Pignoni and Zipursky, 1997). This method was used to generate fluorescently marked, mutant FC clones adjacent to unmarked WT cells, allowing direct phenotypic comparison. Although light microscopy studies of such cells are easy and informative for certain phenotypes (e.g., polarity defects; Goode and Perrimon, 1997; Bilder et al., 2000; Lu and Bilder, 2005), ultrastructural analysis of mutant clones poses the challenge to distinguish the cells of interest from the neighboring ones in the epithelium. We therefore applied our targeting method to acquire FIB-SEM volumes containing fluorescent clones together with a few adjacent unmarked WT cells (Fig. S4).

To this aim, we generated FC clones marked by the expression of CD8-mCherry in which cytoplasmic dynein (*Dhc64C*, hereafter called *Dhc*) was knocked down by transgenic RNAi (see Materials and methods). Cytoplasmic dynein is essential in *Drosophila* epithelia for apical RNA localization, establishment/maintenance of apical-basal polarity, and biogenesis of microvilli by the apical targeting of Cadherin 99C (Wilkie and Davis, 2001; Swan et al., 1999; Horne-Badovinac and Bilder, 2008; D'Alterio et al., 2005; Schlichting et al., 2006). Our ultrastructural analysis showed that *Dhc* RNAi clones marked by CD8-mCherry expression in stage 10 egg chambers displayed all previously described defects associated with the lack of cytoplasmic dynein, including reduced microvillar length (Fig. 6, A, F, and G) and aberrant funnel-like cell shape (Fig. 6, A and N; and Video 4). Interestingly, we found that the decreased length of microvilli was associated with reduced apical deposition of vitelline material, which failed to coalesce and form an even layer of vitelline membrane, as seen in the intercellular space that lies apical to WT FCs (Fig. 6 A and Video 4). Although the presence of an uneven layer of vitelline membrane was previously described in *Cad99C* mutants (Schlichting et al., 2006), it was linked to a failure of microvilli to correctly coalesce vitelline

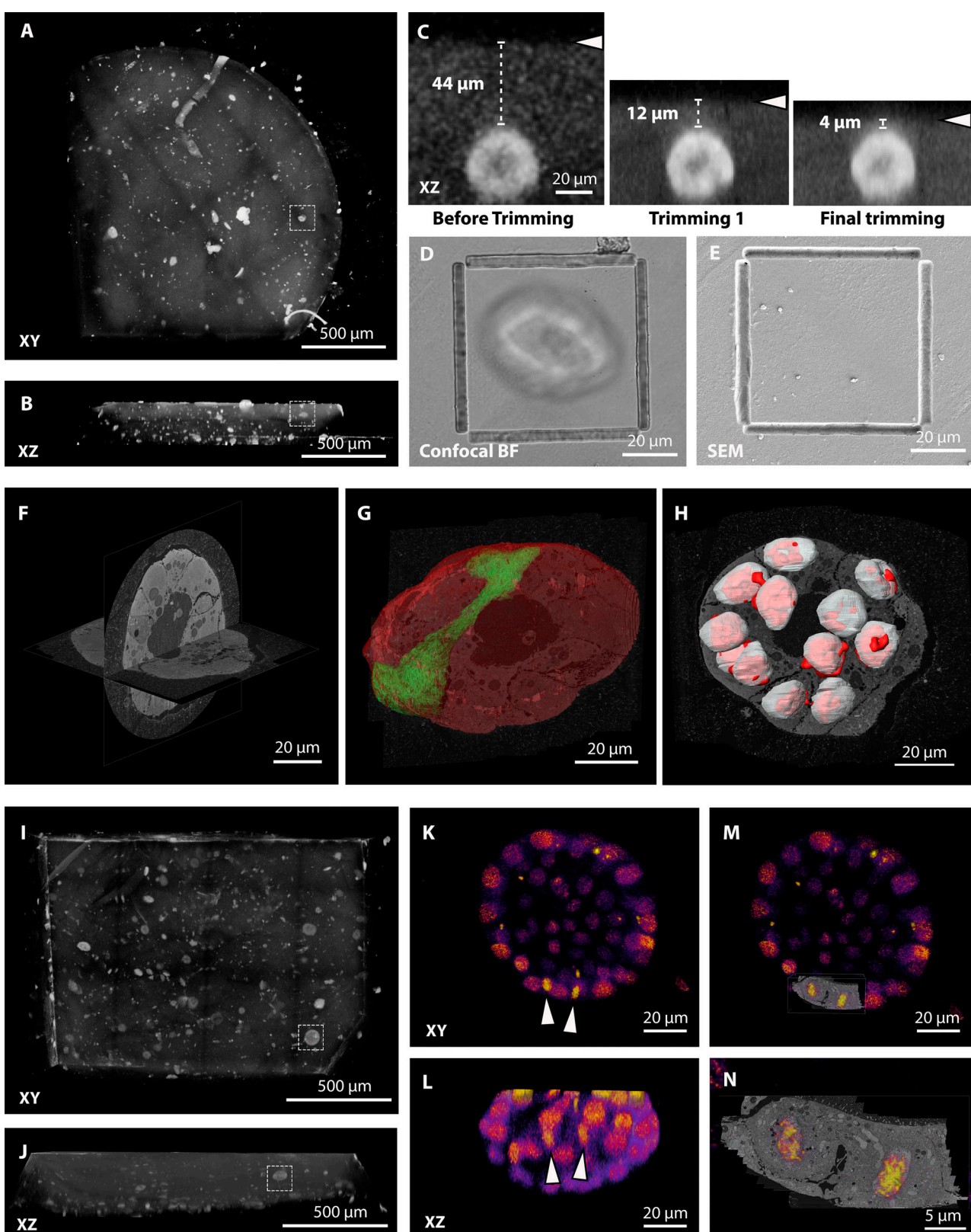

Figure 4. **Targeting of H2B-mCherry–expressing mammary gland organoids grown in Matrigel. (A and B)** Tiled Z-stack confocal acquisition of the resin block. The gray-scale image shows a 3D rendering of the mCherry signal. The autofluorescence allows the identification of the block edges and surface. XY and XZ projection views of the volume are shown in A and B, respectively. The dashed box indicates the organoid of interest. **(C)** XZ views of the organoid in A and its distance to the block surface (arrowheads) during iterative imaging/trimming cycles to approach the ROI in Z. **(D)** Confocal bright field (BF) image of the block surface after laser branding. **(E)** SEM image of the block surface showing the laser mark. Note that the biological sample is not yet exposed at the surface,

making the branding the only reference to target FIB-SEM acquisition. **(F)** FIB-SEM acquisition of the full organoid, achieved at 15 nm × 15 nm × 20 nm voxel size. Orthogonal slices through the volume are shown. **(G)** Segmentation of the organoid (in red) and of a representative single cell (green). At this stage of the organoid development, the cells acquire a complex organization. The cell highlighted has contact to the Matrigel on two sides and forms the lumen with a lateral portion of its protrusion. **(H)** Overlay of the nuclei segmented from the EM volume (white) and from the fluorescence stack (red) shows precise alignment of the datasets, allowing single-cell identification. **(I–N)** Targeting of a mitotic telophase event in an organoid. I and J show the targeting of an organoid (dashed box). After exposing the organoid, a mitotic event could be identified (arrowheads in K and L), and these cells were then targeted for FIB-SEM acquisition at high resolution. **(M and N)** Overlay between the fluorescence dataset and a slice of the FIB-SEM volume.

bodies into vitelline membrane. However, we noticed that *Dhc* RNAi cells also display an accumulation of electron-scattering material, resembling vitelline material, in the basolateral extracellular space (Fig. 6, A–E; and Video 4). Taken together, these results suggest that the defects observed in the apical membrane of *Dhc* RNAi cells might result from a combination of aberrant formation of microvilli and mistargeting of vitelline material to the basolateral domain. Cytoplasmic dynein has also been described to regulate trafficking of the endo-lysosomal system (Reck-Peterson et al., 2018). In several cell types, dynein regulates microtubule minus-end trafficking of late endosomes/lysosomes in coordination with the kinesin motor protein, which directs microtubule plus-end transport of the vesicles (Cabukusta and Neefjes, 2018). In control cells, we observed an apical localization of MVBs (Fig. 6, H, M, and O). In contrast, in *Dhc* RNAi cells, MVBs clustered close to the basal membrane (Fig. 6, H–O; and Video 4), indicating the requirement of dynein for their apical localization in WT FCs. In *Dhc* RNAi cells, the localization of MVB basally, where microtubule plus ends are enriched (Bacallao et al., 1989; Clark et al., 1997), suggests a role

of the plus end–directed kinesin motor in their movement. Further studies will be required to investigate this phenomenon. Another interesting observation was that while WT FCs display a subpopulation of large MVBs in proximity to the apical membrane (Fig. 6, J, M, and P), in *Dhc* RNAi cells the MVBs have a smaller and more homogeneous size (Fig. 6, K, L, N, and P). This is consistent with an effect of dynein on MVB size, potentially through transport of smaller MVBs to the apical side, where they may be able to fuse to form larger structures.

## Discussion

In this paper, we present an easy and reliable workflow to target FIB-SEM volume imaging. We have targeted two single cells in a pellet expressing H2B-mCherry or H2B-mEGFP, three complete H2B-mCherry–labeled organoids, two single mitotic events within an organoid, eight *Drosophila* tracheal terminal cells, and six clones of *Drosophila* ovarian FCs knocked down for dynein heavy chain, with 100% success rate. Compared with targeting methods based on morphological and anatomical cues, this

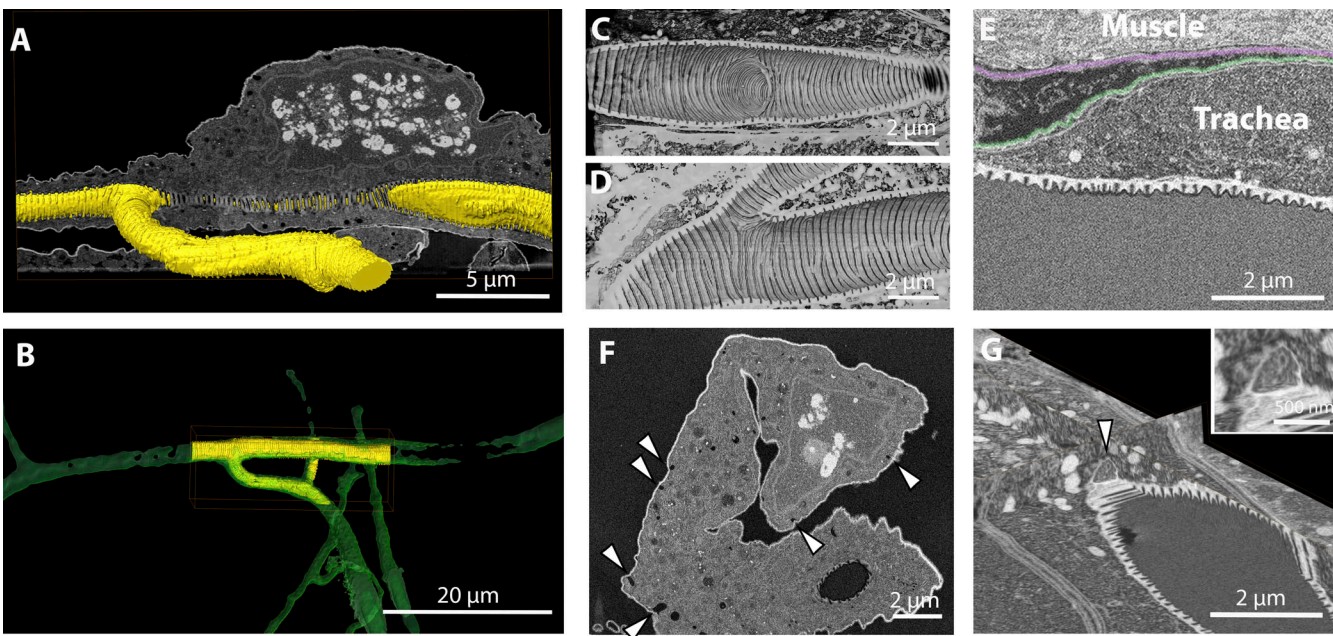

Figure 5. **Imaging of the terminal cell of the trachea in *Drosophila* larva. (A)** Segmentation of the hollow space inside the tracheal tube (yellow) visualized with a raw image of the FIB-SEM acquisition (gray scale). **(B)** Overlay of the segmentation of the tracheal tube obtained from the confocal dataset (green) and from the FIB-SEM data (yellow). **(C and D)** Volume rendering of the inside of the tracheal tube, showing the aECM structures formed in sites of tube branching. **(E)** FIB-SEM image showing the organization of the basal membranes surrounding the muscle cells (segmentation shown in transparency in magenta) and the tracheal cell (segmented in green). **(F)** FIB-SEM image showing a cross section of a tracheal cell. Arrowheads point at invaginations of the basal plasma membrane consistent with membrane trafficking activity. **(G)** 3D visualization of a putative site of fusion of a carrier vesicle containing electron-scattering structures (arrowhead) with the apical plasma membrane of a trachea cell.

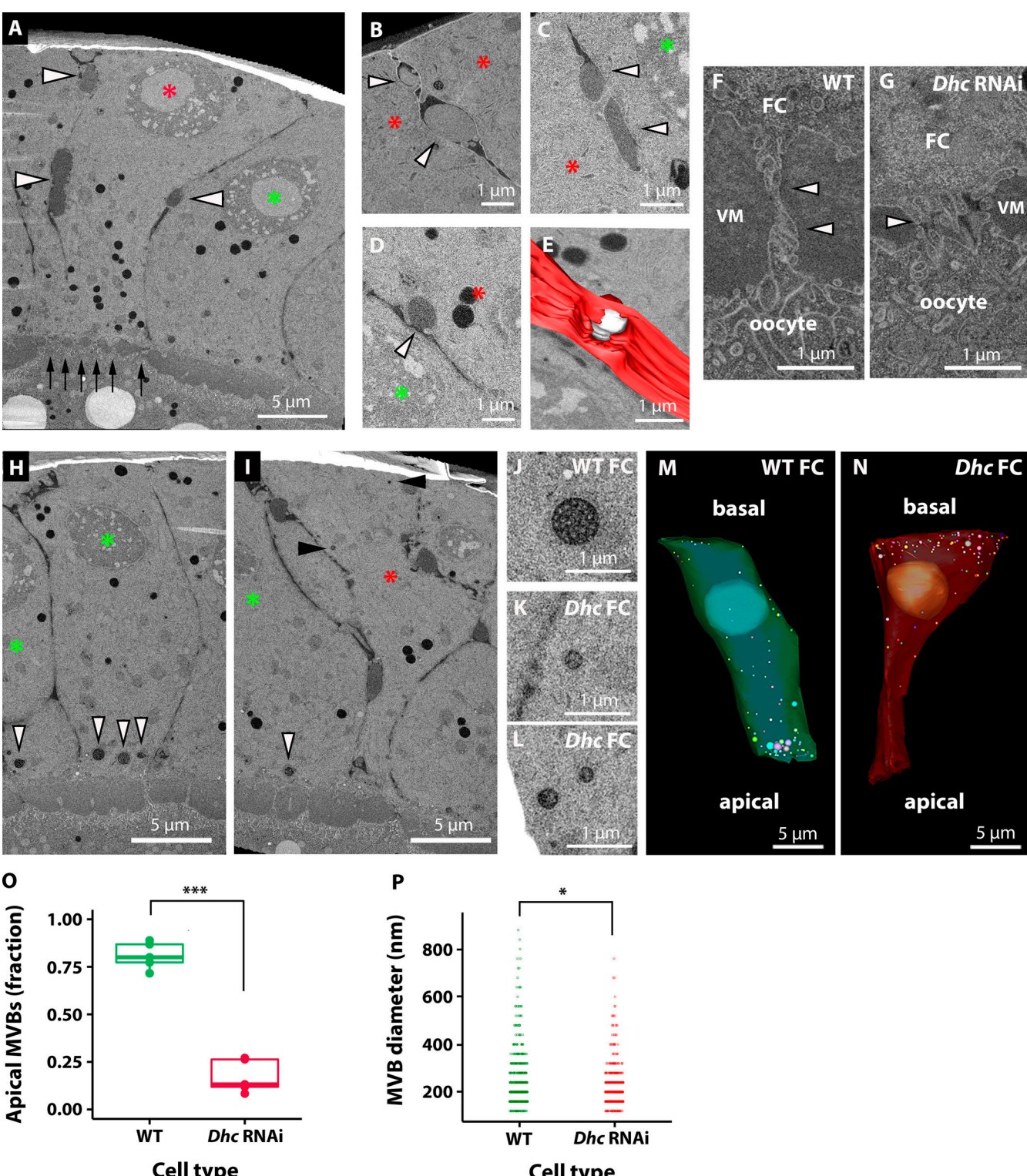

Figure 6. **Imaging of *Dhc* KD cells in the follicular epithelium of *Drosophila* ovaries.** In all panels, red asterisks mark KD cells and green asterisks mark WT cells. **(A)** Overview of the epithelium. Black arrows highlight the reduced space between the oocyte and the KD cell, compared with the neighboring WT epithelium. White arrowheads point to sites of lateral deposition of vitellin membrane–like electron-scattering material. **(B and C)** Higher magnification of material deposited on the lateral side between two KD cells. The material forms an electron-scattering drop (arrowheads) that does not mix with the surrounding extracellular space, which appears darker. **(D)** Potential exocytosis event of electron-scattering material on the lateral side of a KD cell (arrowhead). **(E)** Manual segmentation of the event in D. Plasma membrane of the KD cell is shown in red and the drop of material secreted in white. **(F and G)** High-magnification details of the microvilli between the oocyte and FCs, as indicated in A by black arrows at lower magnification. VM, vitelline material between FCs and oocyte. Arrowheads indicate the microvilli. **(H–P)** Characterization of MVBs. A WT cell accumulates large MVBs on its apical side (H, white arrowheads), whereas in KD cells the MVBs are mostly localized toward the basal side (I, black arrowheads). J shows a typical example of an apical MVB in WT FC. K and L

show basolateral MVBs in KD cells. M and N show segmentation of MVBs in a WT (green) and a KD (red) cell, respectively. The funnel-like shape of the KD cells is also particularly evident in this example. In O, a box plot of the distribution of the MVBs in WT ($n$ = 5) and KD ($n$ = 5) cells. Each point in the plot represents the fraction of apical MVBs in a cell. The plot shows the median and the first and third quartiles of the distribution. Independent $t$ test was used to compare means. ***, P = 2E−6. **(P)** The plot shows the size distribution of MVBs in WT and KD FCs. 458 MVBs from five WT cells and 616 MVBs from five KD cells were pooled. Two-sample Kolmogorov-Smirnov test confirmed that the two distributions are different (*, P = 3.06E-4).

workflow has the advantage of directly using the molecular identity provided by fluorescent labeling of the cells of interest. This has been previously achievable only by integrating pre-embedding fluorescence and FIB-SEM imaging. Such correlation can be very challenging due to anisotropic shrinkage of the samples during EM sample preparation. X-ray microscopic computed tomography can aid in the identification of such distortion, and it has been successfully used to target FIB-SEM volumes (Karreman et al., 2016). However, preserving the fluorescence in the sample and being able to image the very same volume by 3D light and volume SEM techniques has a clear advantage in terms of precision and ease of application. Moreover, our workflow enables targeting of FIB-SEM acquisition of a cell starting from a millimeter-sized EM block in ∼3 h, while Karreman et al. (2016) reported ∼2 d for the x-ray–based targeting. Given the high resolution of fluorescence imaging and the fact that the sample is not altered between the two imaging modalities, we foresee that it will be possible to target even subcellular structures. However, such application would require the development of strategies to align the 3D datasets after acquisition with high accuracy.

The characterization of the mEGFP and mCherry fluorescence in resin shows that our sample preparation causes a considerable fluorescence loss compared with nonembedded samples. We estimate that ∼30% of the original mCherry signal could be detected, whereas only ∼5% of the mEGFP signal was retained. Due to UA autofluorescence in the green channel, mEGFP also has a much lower signal/background ratio in embedded samples compared with mCherry. These measurements provide a reference for the initial fluorescence intensity of the samples required for this approach. We additionally determined that hydration increases fluorescence emission, consistent with previous findings (Peddie et al., 2017). Interestingly, the effect of water increases with incubation time. As these measurements were taken at the block surface and the two FPs display different kinetics, we think that this reflects intrinsic properties of the fluorophores in resin, and it may also have implications for on-section CLEM experiments. Although less hydrophobic compared with epoxy resins, Lowicryl HM20 is nonpolar and therefore not very water permeable. As a consequence, the hydration effect on the fluorophores is limited to a few microns from the surface. At higher depths, the fluorescence drops to a plateau, equivalent to the emission level of nonhydrated FPs. While this level was enough for identification of targets expressing red FPs, it was not sufficient for mEGFP. Altogether, our fluorescence characterization highlights the importance of the fluorophore choice for 3D CLEM experiments.

Our work shows that sample preparation protocols compatible with fluorescence preservation are satisfactory for FIB-SEM milling and imaging. In agreement with a previous report (Porrati et al., 2019), we achieved good contrast in the presence of 0.1% UA and complete absence of osmium. The stability and milling properties of Lowicryl HM20 under the FIB were comparable to the commonly used epoxy resins.

In summary, our data show a reliable workflow for targeting single cells within a large 3D volume based on their molecular signature. This method has enabled us to characterize in a short time specific single cells within a homogeneous epithelium or in a complex tissue. We therefore believe that such a workflow provides many EM laboratories unprecedented options to study cell and developmental biology in 3D.

## Materials and methods

### Cell lines and cell culture

HeLa Kyoto cells stably expressing H2B-mEGFP (Neumann et al., 2010; CLS cat #300673) or H2B-mCherry (Neumann et al., 2010; Euroscarf cat #P30632) were a kind gift of the Ellenberg laboratory (European Molecular Biology Laboratory). They were cultured in DMEM (Gibco) supplemented with 10% fetal bovine serum (Gibco), penicillin/streptomycin (Gibco), and L-glutamine (Sigma). Geneticin or puromycin was added to the culture media of H2B-GFP and H2B-mCherry cells, respectively, for selective pressure. Before high-pressure freezing, the cells were trypsinized and the two cultures were mixed and concentrated by centrifugation (3 min at 194 rcf). The pellet was resuspended in ∼300 µl medium, and 1.5 µl of the suspension was pipetted into the high-pressure freezing carrier.

### Organoid culture

To establish the mouse strain line TetO-MYC/TetO-Neu/MMTV-rtTA/R26-H2B-mCherry, mouse lines TetO-MYC/MMTV-rtTA (D'Cruz et al., 2001), TetO-Neu/MMTV-rtTA (Moody et al., 2002), and R26-H2B-mCherry (Abe et al., 2011; RIKEN; CDB0239K) were crossed into FVB (Friend virus B) background. Housing and care of all animals used in this study were performed at the Laboratory Animal Resources facility at European Molecular Biology Laboratory Heidelberg according to guidelines and standards of the Federation of European Laboratory Animal Science Association. All mice were bred and maintained in a 12-h light/12-h dark cycle, with constant atmospheric conditions (23 ± 1°C temperature; 60 ± 8% humidity) and permanent access to food and water. For establishment of 3D organoids, mammary glands from 8-wk virgin TetO-MYC/TetO-Neu/MMTV-rtTA/R26-H2B-mCherry female mice were dissected and digested following the published protocol (Jechlinger et al., 2009). Single cells were seeded after mixing them with a cold combination of Matrigel Growth Factors Reduced (Corning; 356231), Rat Collagen I (RnD Systems; 3447–020-01), and PBS. Gels were allowed to polymerize before supplying them with Mammary Epithelial Cell Growth Medium

(Promocell, c-21010 and supplement with Mammary Epithelial Cell Growth Supplement [Sciencell; 7652]). Cultures were maintained for 3–10 d in culture in a humidified atmosphere with 5% $CO_2$.

### Drosophila tracheal cell dissection

The Drosophila line used was reported previously (Best and Leptin, 2020) and carries a recombined btl-Gal4 (Shiga et al., 1996) and UAS-DsRed1 (BDSC 6282) element on the third chromosome, driving expression of DsRed in all tracheal cells. The flies were grown on standard cornmeal-agar medium at 25°C. Wandering third-instar larvae were gently collected from the vial wall using a brush and transferred to a droplet of 4°C Shields and Sang medium on a dissection plate. The larvae were filleted according to standard protocol, exposing the dorsal tracheal system attached to the skin, with all internal organs removed. After confirming that the tissue was still alive by observing the twitching of muscles on the skin, we removed the head and posterior end of the fillet. The remaining sample usually contained completely the five segments A1–A5 (corresponding to tracheal dorsal branch pairs 3–7). This was transferred directly to the high-pressure freezing carrier, prefilled with 20% Ficoll (Sigma; PM70) in Shields and Sang medium.

### Heat-shock treatment and Drosophila ovary dissection

The UAS-Gal4 FLP-out system was used to generate marked mutant clones in a WT background (Pignoni and Zipursky, 1997). Flies homozygous for an allele carrying a Gal4-inducible promoter (UAS) upstream of Dhc64C hairpin RNA (BDSC #36698; inverted repeat sequences from https://fgr.hms.harvard.edu/: sense: 5′-CCGAGACATTGTGAAGAAGAA-3′; antisense: 5′-TTCTTCTTC ACAATGTCTCGG-3′) were crossed with hsFlp; arm>f+>Gal4; UAS-CD8-mCherry (kind gift from Juan Manuel Gomez-Elliff, Leptin lab, European Molecular Biology Laboratory, Heidelberg, Germany). The protocol described in González-Reyes and St Johnston (1998) was followed to generate FC clones. Briefly, freshly eclosed females resulting from each cross were collected and mated with w1118 males for 24 h at 25°C on food supplemented with yeast. Flies were heat-shocked for 1 h in a water bath at 37°C, then kept at 25°C with males on yeast. Ovaries were dissected in PBS 39 h after heat shock and immediately high-pressure frozen using 20% Ficoll in Schneider's medium as cryoprotectant.

### EM sample preparation

All samples were high-pressure frozen in their respective freezing media with an HPM010 (AbraFluid), using 3-mm-wide, 200-μm-deep aluminum planchettes (Wohlwend GmbH). FS and resin embedding were performed as described in the Results in an automated AFS2 machine (Leica), using the freeze substitution processor unit. To facilitate the infiltration of Lowicryl HM20 (Polysciences Inc), the temperature was gradually raised to –25°C while increasing the resin concentration in acetone. Finally, the samples were UV polymerized at –25°C. Details are shown in Table S1.

### Confocal microscopy and laser branding

For confocal imaging, the blocks were mounted in a 35-mm glass-bottom culture dish (MatTek) immersed in a drop of deionized water. Fluorescence imaging of all the samples was done using an inverted Zeiss LSM 780 NLO microscope equipped with a 25×/0.8 multi-immersion objective lens (Zeiss, LD LCI Plan-Apochromat 25×/0.8 Imm Korr DIC M27). First, we imaged the whole block using the "Tile Scan" function together with the "Z-Stack" function of ZEN-black. Second, from single targets we recorded high-resolution stacks to assess the overall morphology of the cells of interest. From the acquired Z-stacks, we estimated the relative distance of the target to the surface of the resin block. Next, we trimmed the block using an ultramicrotome (Leica UC7) to remove excessive resin material on top of the cells of interest and iteratively repeated the imaging and trimming steps until the selected cell was positioned at a distance of ~1–5 μm to the resin block surface. Branding of the resin surface was done using the two-photon Coherent Chameleon Ultra II Laser of the Zeiss LSM 780 NLO microscope. To engrave a rectangular region on the resin block surface, we used the "Bleaching" function of ZEN black. Different settings were tested to find the optimal parameters for branding. We obtained the best results applying ~117 mW at the sample plane of the two-photon laser running at 765-nm wavelength with an effective scan speed of 2.55 μs over 50 iterations in a selected small rectangular region of the block surface. However, the success of branding and the energy level required were variable between samples, different areas on one resin block, and different sizes of branding regions. We therefore always started with minimal laser intensities and repeated the branding scan with increasing intensity until a clear brand appeared in the transmitted light image of the confocal.

### H2B-mEGFP and H2B-mCherry fluorescence characterization in block

A mixed suspension of HeLa cells expressing H2B-mCherry or H2B-mEGFP was prepared and embedded. Imaging was performed with a Zeiss LSM780 NLO (see above). To find the fluorescence loss during sample preparation, we compared cells in block with the same cell lines suspended in Matrigel and fixed with 4% PFA (EMS). We determined the laser intensity needed to produce images with similar dynamic range for the two conditions, while keeping other acquisition parameters (dwell time, photomultiplier gain) unchanged. We used a laser intensity of 0.35% and 0.5% for nonembedded mEGFP and mCherry and 7% and 1.5% for resin-embedded mEGFP and mCherry, respectively.

For the measurement of the hydration effect on fluorescence emission (Fig. 2, A and B), we placed the block in a Mattek dish without water and acquired a Z-stack with 2-μm Z increment and the settings as described above. After that, a drop of water was added between the glass and the block surface, the same FOV was retrieved, and Z-stacks with the same settings were acquired at 2, 30, 60, and 90 min. For the measurements, the confocal plane through the block surface was considered. In each confocal plane, both cell lines were visible so that the behavior of the two FPs could be compared.

For the measurements of the fluorescence decay as a function of distance from the block surface (Fig. 2, C and D), we left the block in water for 90 min, and we acquired a confocal Z-stack as described above.

For the photobleaching measurement, a FOV containing both cell lines was consecutively scanned 250 times with each laser line, with the same settings used for standard imaging.

**FIB-SEM acquisition**

After confocal imaging and branding for targeting, the blocks were mounted on an SEM stub using silver conductive epoxy resin (Ted Pella). In case the block was later to be imaged again at the confocal to identify a second target, instead of the silver epoxy (which requires polymerization at 60°C), we attached the block to the stub using a mixture of super glue (Loctite) and colloidal silver liquid (Ted Pella). After mounting, the blocks were gold sputtered with a Quorum Q150R S coater.

For FIB-SEM acquisitions, we used a Zeiss CrossBeam XB540 or XB550, using the Atlas3D workflow (Fibics Inc.). Briefly, a platinum coat (~1 μm thick) was deposited over the area marked by laser branding. Autotuning marks were milled on the platinum surface and highlighted with carbon. We milled large trenches with 30-kV FIB beam acceleration voltage and 30-nA current and polished the surface with 7- or 15-nA currents. Precise milling during the run was achieved with currents of either 700 pA or 1.5 nA. For all experiments, the SEM imaging was done with an acceleration voltage of 1.5 kV and current of 700 pA, using a backscattered electron (ESB) detector. Pixel sizes and dwell times were different depending on the volume that we acquired. For relatively small, high-resolution volumes, we acquired at 8-nm (single cells in organoids) or at 10-nm isotropic voxel size (*Drosophila* ovarian FCs, *Drosophila* trachea). For very large volumes (entire organoids—spheroids up to 65 μm in diameter), we acquired at 15 nm × 15 nm × 15/20 nm voxel size. Dwell times ranged between 8 and 12 μs but were occasionally increased to 20 μs during the run to obtain single images with high signal-to-noise ratio.

**Image processing, dataset registration, visualization, and segmentation**

FIB-SEM image stacks were aligned using either the "Linear stack alignment with SIFT" plugin in Fiji or the "Alignment to median smoothed template" workflow recently described (Hennies et al., 2020).

For visualization and registration of the different imaging modalities, we used Amira (version 2019.3 or 2020.1; Thermo Fisher Scientific). The volumes of the confocal and FIB-SEM datasets were aligned using the transformation editor. Because of the geometry of the beams in a FIB-SEM, the EM volume is orthogonal to the confocal volume in all cases. Therefore, a 90° rotation of one of the two volumes along the X or Y axis was necessary to register the two imaging modalities. The transformation was then further refined. When the grooves left by the laser branding were visible in the FIB-SEM volume, they were used as landmarks for the registration. Otherwise, morphological features of the target cells were used (e.g., branching points of the tracheal tubes or characteristic shapes of organoids and *Drosophila* FCs). Videos were made using Amira (version 2020.1).

**MVB quantification**

In WT or *Dhc64CRNAi* FCs, MVBs were segmented using Imod version 4.10.43 (Kremer et al., 1996). MVBs were identified as structures with a clear lumen containing one or more small vesicles of homogeneous size and segmented as spherical "scattered objects" of the corresponding diameter. To assess their distribution (Fig. 6 O), we split the cell into two domains with a plane perpendicular to the apical-basal axis going through the center of the nucleus and quantified the fraction of MVBs present in each side. For their size distribution (Fig. 6 P), all MVB sizes for each cell type were pooled. The statistical analysis was performed as follows. The assumption of normality was tested using the Shapiro Wilk test, with a P value of 3.517401e-24 and 2.520202e-28 for the WT and knockdown (KD) distributions, respectively, which indicates that distributions are not normal. Therefore, a Two-Sample Kolmogorov-Smirnov test was used, which gave a statistical distance = 1.291423e-01 and P value = 3.059291e-04 (calculated critical value is 0.121081017606705 for 468 MVBs analyzed in WT cells and 616 for KD cells, α = 0.001). Therefore, the two samples come from populations with a different distribution.

**Online supplemental material**

Fig. S1 shows the fluorescence characterization of H2B-mCherry–expressing primary mouse mammary gland organoids. Fig. S2 shows the workflow to sequentially target multiple fluorescent ROIs at different depths in the block. Fig. S3 illustrates the targeting of a *Drosophila* trachea terminal cell. Fig. S4 shows the targeting of a fluorescent clone of *Drosophila* ovarian FCs and an overlay of the fluorescence with the FIB-SEM segmentation. Video 1 and Video 2 show examples of the workflow from the confocal volume of the block to the segmentation of the FIB-SEM volume and its overlay with the fluorescence dataset. Video 1 illustrates the targeting of an entire mammary gland organoid, and Video 2 depicts a single mitotic event within an organoid. Video 3 and Video 4 highlight the main findings of the ultrastructural characterization of the *Drosophila* trachea terminal cell and of the ovarian FC clones, respectively. Table S1 is a detailed FS protocol used in the workflow.

## Acknowledgments

We thank the Electron Microscopy Core Facility, the Advanced Light Microscopy Facility, and the animal facility at European Molecular Biology Laboratory for their support. We thank the Transgenic RNAi Project at Harvard Medical School (National Institutes of Health/National Institute of General Medical Sciences R01-GM084947) for providing transgenic RNAi fly stocks. We are grateful to Dr. José Serra Lleti for his help with the statistical analyses and to Dr. Alessandra Reversi for critically reading the manuscript.

This work was supported by the European Molecular Biology Laboratory. L. Cassella was supported by Deutsche Forschungsgemeinschaft (Germany) DFG-FOR 2333 grants EP 37/2-1 and EP 37/4-1 to A. Ephrussi. E. D'Imprima was supported by a fellowship from the European Molecular Biology Laboratory interdisciplinary postdoctoral program under Marie Skłodowska-Curie Actions COFUND (EI4POD). J. Mahamid acknowledges funding from the European Molecular Biology Laboratory and the European Research Council (ERC 3DCellPhase– 760067).

The authors declare no competing financial interests.

Author contributions: P. Ronchi conceptualized the study. P. Ronchi and P. Machado developed the method with the help of S. Schnorrenberg. P. Ronchi, P. Machado, E. D'Imprima, B.T. Best, L. Cassella, and M.G. Montero performed the experiments. P. Ronchi, P. Machado, and G. Mizzon performed data analysis and visualization. P. Ronchi, B.T. Best, and L. Cassella wrote the manuscript. M. Jechlinger, A. Ephrussi, M. Leptin, J. Mahamid, and Y. Schwab supervised the project. All the authors reviewed the manuscript.

Submitted: 16 April 2021

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

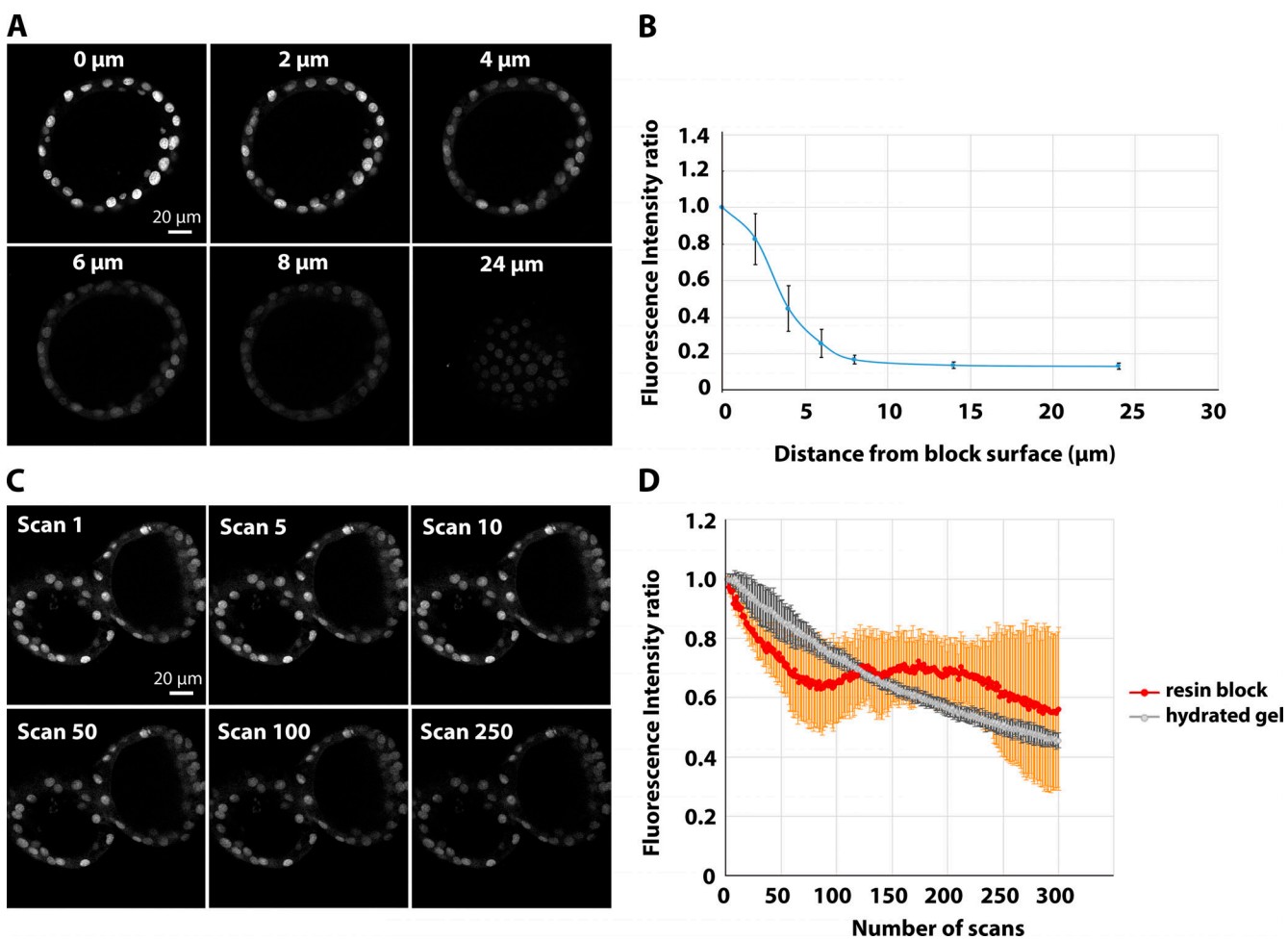

Figure S1.　**Characterization of the behavior of mCherry fluorescence in organoids in resin block. (A and B)** Fluorescence intensity measurements of H2B-mCherry in mammary gland organoids. Confocal stack of a representative sample in A. The integrated fluorescence intensity of 10 nuclei per confocal slice was measured, and the average (±SD) is plotted in B in relation to the distance from the block surface. The data are normalized to the average fluorescence intensity of the nuclei at the surface. **(C and D)** Bleaching curve of H2B-mCherry in a resin block. A FOV containing organoids was consecutively scanned 250 times, with the settings described in the Materials and methods section. Images are shown in C, and quantification of the integrated fluorescence intensity of nuclei from three independent organoids (±SD) is shown in D (red curve). Each experimental point in the chart is represented as a fraction of the fluorescence intensity in the first image. The bleaching experiment of H2B-mCherry in PFA-fixed hydrated Matrigel was conducted with the same settings used for resin-embedded organoids. An average of three independent organoids (±SD) is represented in gray.

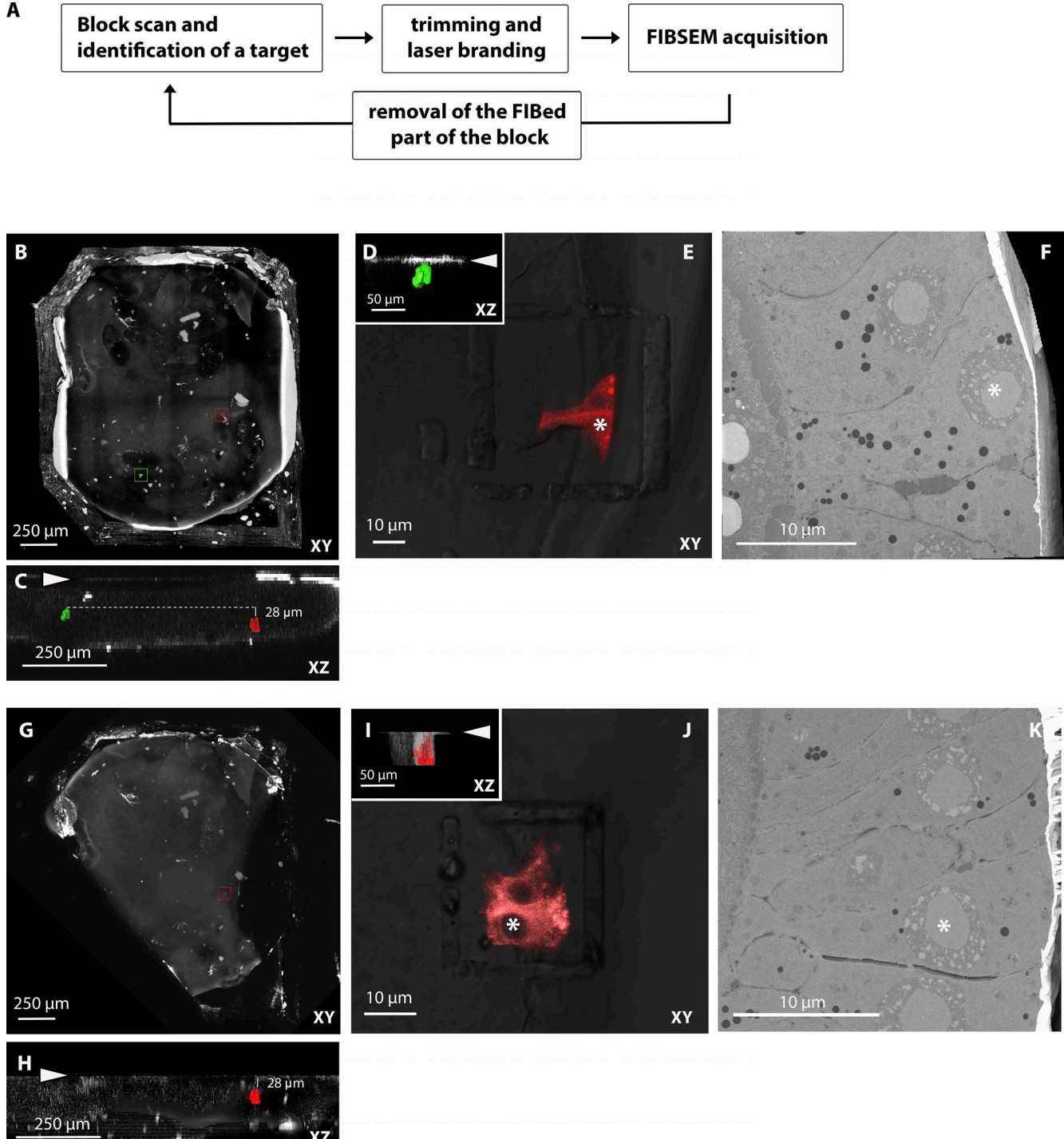

Figure S2. **Targeting of multiple cells in the same block (*Dhc* KD cells in the follicular epithelium of *Drosophila* ovaries expressing CD8-mCherry).** **(A)** Workflow. **(B)** Tiled Z-stack scan of the entire resin block in XY view, with the two targets highlighted in green and red. **(C)** XZ view of the same block with the two fluorescent targets segmented in the same colors. The block surface before trimming is indicated by the arrowhead. The red target is 28 μm deeper than the green one. Therefore, the two cells have to be exposed sequentially for FIB-SEM acquisition. **(D)** Z targeting of the first group of CD8-mCherry–positive cells (segmented in green). The image shows the position of the fluorescent cell (segmented) and the block surface (arrowhead) after trimming. **(E)** Confocal imaging of the block after trimming and laser branding around the first cells of interest (XY view). **(F)** FIB-SEM acquisition of the same volume. The asterisks in E and F indicate the same cell viewed from orthogonal orientations. **(G and H)** Tiled Z-stack scan of the same resin block after removing the area imaged by FIB-SEM. Note that the lower left corner of the block, previously containing the green target, is missing. The XZ view in H shows that the second target (red) is now 28 μm deep from the block surface (arrowhead). **(I)** Z targeting of the second group of cells (segmented in red). The image shows the position of the fluorescent cell (segmented) and the block surface (arrowhead) after the second trimming. **(J)** Confocal imaging of the block after trimming and laser branding around the second cells of interest (XY view). **(K)** FIB-SEM acquisition of the same volume. The asterisks in J and K indicate the same cell.

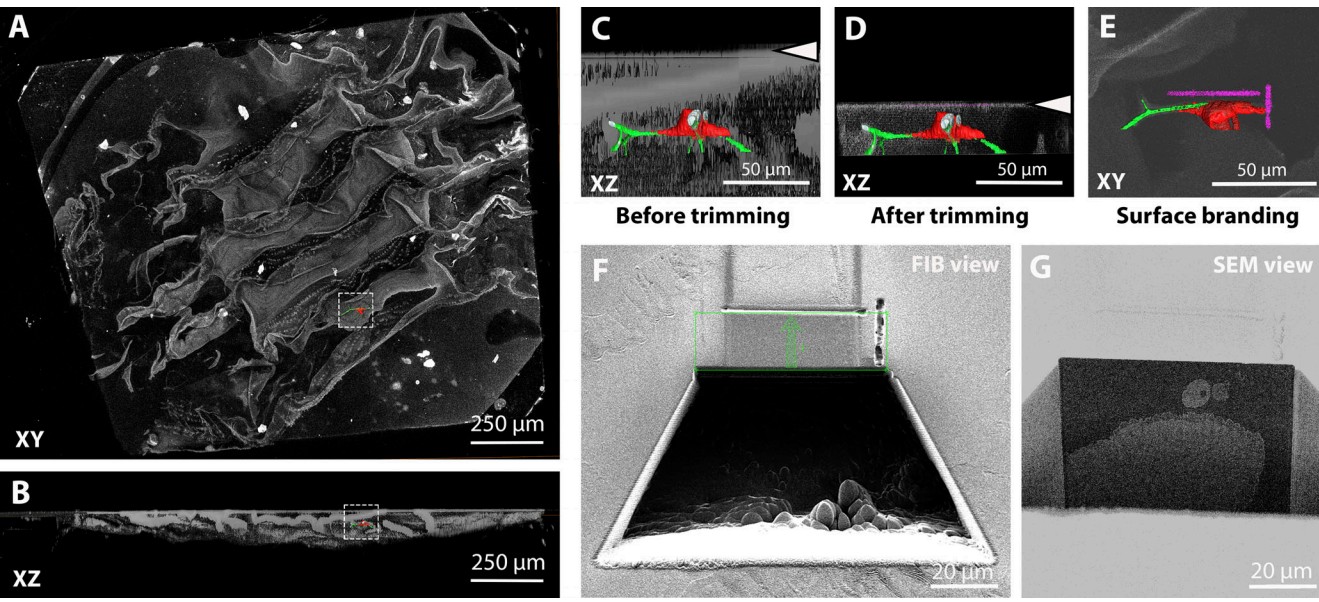

Figure S3.  **Targeting of tracheal terminal cell in *Drosophila* larva. (A and B)** Tiled confocal Z-scan covering the entire volume of the block containing tissue. The gray-scale image shows a volume rendering of the thresholded autofluorescence signal (green channel) of the block (clearly visible is the autofluorescence of the epidermal cuticle) in XY (A) or XZ (B) views. The segmented cell of interest (in red, the DsRed fluorescence of the cell; in green, autofluorescence of the ECM of the tracheal tube) is highlighted in the boxed area to visualize its location in the block. **(C and D)** In gray scale, XZ view of the volume rendering of the fluorescence of the block, with the cell of interest segmented. The arrowheads indicate the position of the block surface before (C) and after (D) trimming. **(E)** XY view of the same volume that shows the two-photon branding of the block surface marking the ROI to be acquired by FIB-SEM (segmented in magenta). **(F and G)** Images of the block during FIB-SEM run setup: 50-pA FIB image acquired with secondary electron detector (F) and 1.5-keV 700-pA SEM image acquired with ESB detector (G). In F, the green box delimits the area that will be milled during the FIB-SEM experiment, and the green arrow shows the milling direction.

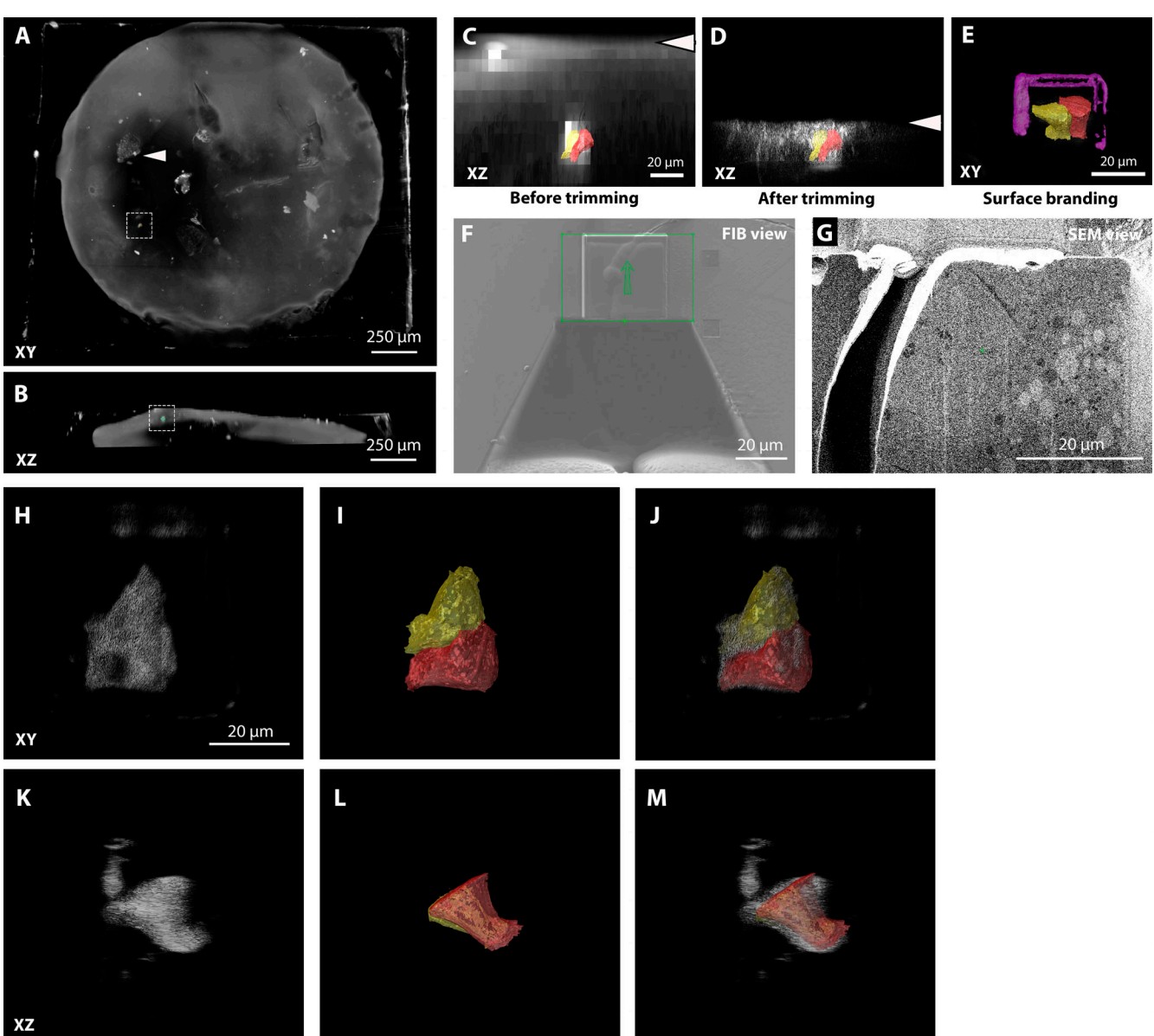

Figure S4. **Targeting of a clone of *Drosophila* ovary FCs expressing *Dhc* RNAi and CD8-mCherry. (A and B)** Tiled Z-stack confocal acquisition of the resin block. The gray-scale image shows a rendering of the red signal. A few mCherry-expressing cell clones are visible (e.g., arrowhead) as well as the auto-fluorescence background of the Ficoll-containing freezing medium. The target of the FIB-SEM acquisition is highlighted by the dashed box, and segmentation is shown to facilitate its visualization. A and B show XY and XZ views of the block, respectively. **(C and D)** In gray scale, XZ view of the volume rendering of the fluorescence of the block, with the cells of interest segmented in two different colors. The arrowheads show the position of the block surface before (C) and after (D) trimming. **(E)** XY view of the same volume that shows the two-photon branding of the block surface limiting the ROI to be acquired by FIB-SEM (magenta segmentation). **(F and G)** Images of the block during FIB-SEM run setup: 50-pA FIB image acquired with secondary electron detector (F) and 1.5-keV 700-pA SEM image acquired with ESB detector. A crack at the interface between the basal membrane of the ovary and the empty resin is visible here. This happened frequently during the branding of *Drosophila* oocyte samples but did not affect the structure of the FCs. **(H–M)** Overlays of the fluorescence (H and K) and segmented FIB-SEM (I and L) datasets show the precision of the registration in different orientations.

Video 1. **Targeting of the FIB-SEM acquisition of a full-volume mouse mammary gland organoid.** H2B-mCherry–expressing organoids grown in Matrigel were prepared by high-pressure freezing and FS (as described in the manuscript). The video shows the targeting method from the identification of the target in the full block confocal volume to the final FIB-SEM acquisition. Finally, the overlay of the nuclei segmentation from the confocal (in red) and the FIB-SEM datasets (white) shows the precision of the method. The video plays at 25 frames per second.

Video 2.   **Targeting of a mitotic event in a mammary gland organoid.** A single mitotic event was identified in a millimeter-size mammary gland organoid 3D culture, based on the nuclear H2B-mCherry fluorescence pattern. After precise targeting of the ROI, the volume of the mitotic cells was acquired by FIB-SEM. The overlay of the nuclei segmentation of the two cells of interest in the two imaging modalities proves the efficiency of the targeting. The video plays at 25 frames per second.

Video 3.   **Details of the ultrastructure of the *Drosophila* tracheal terminal cell.** A *Drosophila* larval trachea terminal cell was targeted for FIB-SEM acquisition as described above. The video shows ortho-slices through the volume to highlight interesting biological details (e.g., the structure of the aECM taenidia, the double layer of basal lamina between trachea and muscle cells, and a potential fusion event of a vesicle carrier containing ECM-like material fusing to the apical membrane of the tracheal cell). The video plays at 11 frames per second.

Video 4.   **Details of the ultrastructure of *Drosophila* ovary FCs KD for dynein heavy chain.** A clone of *Drosophila* ovary FCs KD for Dynein heavy chain was targeted thanks to its coexpression with CD8-mCherry. The video shows ortho-slices through the volume to highlight interesting biological details (e.g., reduced space between the apical side of the KD cells and the oocyte with shorter microvilli, mistargeted lateral deposition of vitelline-like material, and MVB basal relocation). The red fluorescence volume is overlaid in transparency to highlight the KD cells. The video plays at 9 frames per second.

**Provided online is Table S1, which summarizes the FS protocol (Leica AFS2 with FSP unit).**

