## [Peer Review File · The Journal of Cell Biology]

High precision targeting workflow for volume electron microscopy

Paolo Ronchi, Giulia Mizzon, Pedro Machado, Edoardo D'Imprima, Benedikt Best, Lucia Cassella, Sebastian Schnorrenberg, Marta Montero, Martin Jechlinger, Anne Ephrussi, Maria Leptin, Julia Mahamid, and Yannick Schwab

Corresponding Author(s): Paolo Ronchi, EMBL and Yannick Schwab, EMBL

Review Timeline:

Submission Date:	2021-04-16
Editorial Decision:	2021-04-27
Revision Received:	2021-05-27

Monitoring Editor: Andres Leschziner

Scientific Editor: Melina Casadio

Transaction Report:

DOI: <https://doi.org/10.1083/jcb.202104069>

Review#1

1. How much time do you estimate the authors will need to complete the suggested revisions:

Estimated time to Complete Revisions (Required)

(Decision Recommendation)

Less than 1 month

2. Evidence, reproducibility and clarity:

Evidence, reproducibility and clarity (Required)

In their manuscript, Ronchi P. et al. present a thorough and very well detailed workflow for 3D correlative-light and electron microscopy of whole cells in large tissues. Their approach of iterative block trimming and fluorescence imaging combined with laser branding allowed them to explore previously inaccessible tissues and questions. They imaged mammary gland organoids, and resolved the organization of the cells in the organoid and mitotic events. They also specifically targeted tracheal terminal cells of a 3rd instar *Drosophila* larvae labeled with cytoplasmic DsRed to study their ultrastructure, and several *Drosophila* ovarian follicular cells (FC) where the cytoplasmic motor protein dynein was knocked down (KD) by RNAi.

In the tracheal cells, they observed connected secretory vesicles, probably delivering extra-cellular matrix to the trachea tube. They also found that the overall shape of dynein KD FCs is distorted compared to WT, and that the localization of multi-vesicular bodies/endosomes inside the FCs changed from an apical to basal membrane localization. Although the approach is not entirely new, the manuscript certainly paves the way for future studies to obtain ultrastructural information from large specimens and combine it with meaningful fluorescence information, it's also beautiful and polished.

Minor comments:

1. The authors state that they (line 145) that they found the optimal concentration of UA and the best compromise between EM contrast and fluorescence preservation. However, no detail is provided as to how

these parameters were experimentally determined.

2. More detail as to how the block face was mounted and kept parallel to

the glass bottom dish would be helpful. Also, what was the optical slice of the confocal and what was the increment in Z?

3. Have the authors tried fluorophores with shorter wavelength (like GFP)? And **if** so, have they estimated the penetration depth in resin? This would be informative because many GFP lines already exist in the *Drosophila* model.

4. In figure 6 **i**, how did the authors identify the structures to be MBVs close to the basal surface in the mutant seeing as they do not look like the MVBs seen in WT cells? Similarly, how were the structures identified as endosomes in figure 5f? Can the authors quantify the total MVBs in the apical/basal membranes from both RNAi KD and WT?

5. The motivation/question in the case of *Drosophila* samples was clear but not so obvious in the case of the mammary gland organoids. **It** would be nice **if** the authors could give a bit more information.

6. In the introduction (line 124). The dimensions are given in microns and millimeters, which can be a bit confusing.

7. In the discussion (lines 427-431), "sample preparation protocols compatible with fluorescence preservation have proven satisfactory for FIB- SEM milling and imaging" have also been shown by others (Porrati et al., 2019).

8. Figure 1:

- **It** would be helpful **if** the cell referred to in **g** was highlighted.

- Is the cell in **(h)** the one in **g** or in **a** as written?

- Is the image in **(k)** inverted compared to **(i)**? Figure 2:

- In panel **d** **it** seems that some numbers on the x axis were duplicated. Figure 5:

- How does the perfect overlap confirm the accuracy of targeting?

- In panel **(e)** **it** was not particularly easy to understand what is the basal lamina.

- In panel **(g)** the fused vesicle is not as clear as the movie. **I** also found **it** open to interpretation whether this is in fact a fused

vesicle.

3. Significance:

Significance (Required)

The increasing demand for volume electron microscopy brings a lot of challenges to correlative light and volume electron microscopy workflows. Although the methods used by the authors are not new, their combination is original. The manuscript will certainly contribute to the field of correlative light and volume EM and provide a rather detailed protocol that can be reproduced by others. The workflow is also more efficient than what was previously achieved using x-ray instead of light microscopy (Bushong et al., 2015; Karreman et al., 2016).

Review#2

1. How much time do you estimate the authors will need to complete the suggested revisions:

Estimated time to Complete Revisions (Required)

(Decision Recommendation)

Less than 1 month

2. Evidence, reproducibility and clarity:

Evidence, reproducibility and clarity (Required)

In this manuscript, Ronchi et al describe a workflow designed to facilitate the identification and downstream relocation of fluorescently tagged regions of interest within millimetre scale samples, ending with focused ion beam SEM acquisition of the target area. The work follows a logical progression, is well thought out, explained, and illustrated, with proof of concept experiments that are followed up by examples of systems where the potential for the application of the workflow in a 'real' biological question is demonstrated.

For me, the title reads better as ...targeting for FIB SEM acquisition...

I have only minor suggestions for the revision of the manuscript from this **initial** version. The introduction, and introductory paragraphs for the two model systems would benefit from some revision to make them more concise however.

****Summary****

Line 22 - omit large.

Introduction

Line 66 - It's probably clearer to discuss this concept as conductivity rather than grounding.

Line 105 - Peddie and Collinson 2014 is not the correct reference for this statement. Presumably this is supposed to be Peddie et al 2014?

Line 124 - The external diameter of the carrier would give 7 mm², but the internal diameter is smaller, so this size is slightly overstated.

Results

General comment - I find the use of NxNxN/N nm³ to be a confusing way of expressing the measurements, so would suggest splitting these up to express as: N nm³ or NxNxN nm.

Line 141 - no water was used in the FS mixture, and so wasn't needed for preservation of fluorescence? Dry/100% acetone? If no water is needed, this detail should be discussed.

Line 142 - could the authors elaborate on the statement about timing and sample types, to give a better understanding of the context.

Line 150 - on the choice of fluorophores, did the authors examine any

shorter wavelengths, or was the decision to use red/far red based on any other evidence? Anecdotally, red and far red fluorophores may offer better preservation and less longevity in this context, but could the authors elaborate on their reasoning behind the choice shown here?

Line 168 - did immersion in water give rise to any distortion of the resin, or is HM20 sufficiently hydrophobic that this was not a concern? Mismatches in refractive indices (resin, water, glass, oil) could also presumably give rise to some small inaccuracies in depth prediction?

Line 169 - was it possible to quantify the increase in signal? If the block is being hydrated, but the block is not absorbing water (re above point), then it must only be surface fluorophores that are hydrated and give rise to this increase in signal?

Line 179 - presumably this is a result of the surface of the block being hydrated (re above points). This is mentioned later, but could be explicitly stated here to make the point more strongly.

Line 188 - Peddie et al 2014 contains some limited data for mCherry in sections that could be worth mentioning in support of the findings of reduced photobleaching rates.

Line 268 - It is not explicitly stated earlier, but multiple targets at similar depths would also be possible, presumably.

Discussion

Line 421 - sections cannot be repeatedly imaged without bleaching too much? Please elaborate on this statement to help strengthen the point as it isn't mentioned earlier in the results.

Line 435 - FIB SEMs and 2Pi systems are not really so 'common' in the sense suggested, so this final statement should be reworded.

M&Ms

Line 540 - grooves, not groves

Figure 3 legend

Overall, it's a workflow comprising many methods, so it's best described as a schematic of the workflow.

Confocal panel - target, not targets, and depth is misspelt.

Figure 4 legend

Line 834 - as far as I can see, this is a different organoid that isn't shown in a and b, so this text should be removed.

Figure 5 legend

Line 840 - was, not is

Figure 6

a) It would help with clarity to also put e.g. white arrows on the WT epithelium

f,g) It isn't really clear on first viewing what these images show, so they would benefit from some labels.

****Minor stylistic comments****

There should be a space between numbers and units; this is inconsistent throughout.

The use of black versus white text on the figures is inconsistent.

Table 1 - is it in the supplementary material or not? **If it is, it** should be referenced as such in the text. The formatting could use some refinement to match the standard of the other figures.

Capitalisation is inconsistent throughout.

3. Significance:

Significance (Required)

The manuscript describes a workflow that connects several pre-existing methods to enable precision targeting of individual fluorescently tagged structures within a larger sample volume. The possibility for multi-modal imaging within a single embed specimen facilitates correlation of data for structure, with that of function. The work will be of interest to all scientists in the field of correlative microscopy.

Review#3

1. How much time do you think the authors will need to complete suggested revisions

Estimated time to Complete Revisions (Required)

(Decision Recommendation)

Between 1 and 3 months

2. Evidence, reproducibility and clarity:

Evidence, reproducibility and clarity (Required)

The manuscript is written very clearly overall. I would like to raise a number of issues that the authors might address. Most are at the level of proof-reading.

The workflow **still** depends on availability of a specialized confocal microscope with two-photon laser excitation for marking the region of interest. A tweak to the method might simply be to scratch or etch markings onto the planed surface near the edges. Provided a motorized stage is available on the light microscope, the region of interest could be located precisely with reference to those, and then relocated in the SEM. **It** would be enough to suggest this, or another similar method, for those who don't have access to the two-photon microscope.

The second is to clarify in the text that the top-down view of the confocal microscopy is orthogonal to that of the FIB. This appears as a note in the caption to Figure 1, but **it** is an important point to align the expectations of readers who are not closely familiar with the methods.

The legend labels in Figure 1 do not match the figure itself, as if it were recompiled from an earlier draft: g-j) refers next to a).

The decrease in fluorescence intensity with depth into the specimen remains a bit ambiguous. The significant part of the text is dedicated to the suggestion that inherent protein fluorescence is affected by water content in the resin. After cutting back from the surface, are the originally deeper layers **still** dim, or do they become brighter? In other words, is the effect chemical or optical? Loss of confocal intensity with depth would be expected on the basis of a refractive index mismatch to the design parameters of the objective, especially for high numerical aperture. The objective is specified as multi-immersion but no further details are given. Another easy test would be to embed fluorescent beads as intensity standards. There

could also be absorption of the fluorescence emission by the resin and stain, but such a strong effect in a few tens of microns would suggest that the block is quite dark. That seems inconsistent with the images in supplementary figures. Personally I was not bothered by the dimming in depth, since the conclusions do not depend on quantitative fluorescence intensities.

In some cases pre-embedding correlative imaging can be quite successful, for example in studies of Jost Enninga (e.g., Mellouk et al, Cell Host & Microbe 2014) or Eric Jorgensen (Watanabe et al, Nature Methods 2011). Do the authors see a distinction between adherent cell cultures and unsupported tissues or tissue sections?

Other investigators have insisted that FIB-SEM requires especially heavy labelling. What was done differently here to make the light labeling possible? Such clues may be very useful to ongoing developments in the literature. Also, the present protocol skips osmium staining entirely. The authors must have compared images with and without osmium. What visible features do we lose as a result?

Perhaps the greatest challenge to large volume electron microscopy is to deal with rare events. Correlative fluorescence light-electron microscopy effectively addresses the issue of finding the region of interest in a two-dimensional specimen such as a thin section or even a monolayer cell culture. For tissues the solutions are **still** at large. It is almost always impractical to image an entire organ at the resolution required to see macromolecules (work of Harald Hess being the exception that proves the rule). The issue is especially acute where the imaging is destructive, as in the case of serial block-face and FIB-SEM tomography. MicroCT has been used so far as the method of choice in the work-up to locate the region of interest within a large specimen, but the approach requires expensive equipment and time-consuming analysis. Furthermore, it can provide directional clues solely on the basis of morphology.

Fluorescence would be a far simpler tool, and more informative when labeling is directed to specific molecular components. The manuscript of Ronchi et al provides a much-needed demonstration and detailed set of instructions for 3D CLEM en route to FIB-SEM volume imaging. The examples are presented are both convincing and esthetic. Success depended on integration of a number of factors, including changes to the specimen preparation, so the workflow will be very useful. In short, I recommend publication.

I would like to raise a number of issues that the authors might address. Most are at the level of proof-reading.

The workflow **still** depends on availability of a specialized confocal microscope with two-photon laser excitation for marking the region of interest. A tweak to the method might simply be to scratch or etch

markings onto the planed surface near the edges. Provided a motorized stage is available on the light microscope, the region of interest could be located precisely with reference to those, and then relocated in the SEM. It would be enough to suggest this, or another similar method, for those who don't have access to the two-photon microscope. The second is to clarify in the text that the top-down view of the confocal microscopy is orthogonal to that of the FIB. This appears as a note in the caption to Figure 1, but it is an important point to align the expectations of readers who are not closely familiar with the methods.

The legend labels in Figure 1 do not match the figure itself, as if it were recompiled from an earlier draft: g-j) refers next to a).

The decrease in fluorescence intensity with depth into the specimen remains a bit ambiguous. The significant part of the text is dedicated to the suggestion that inherent protein fluorescence is affected by water content in the resin. After cutting back from the surface, are the originally deeper layers still dim, or do they become brighter? In other words, is the effect chemical or optical? Loss of confocal intensity with depth would be expected on the basis of a refractive index mismatch to the design parameters of the objective, especially for high numerical aperture. The objective is specified as multi-immersion but no further details are given. Another easy test would be to embed fluorescent beads as intensity standards. There could also be absorption of the fluorescence emission by the resin and stain, but such a strong effect in a few tens of microns would suggest that the block is quite dark. That seems inconsistent with the images in supplementary figures. Personally I was not bothered by the dimming in depth, since the conclusions do not depend on quantitative fluorescence intensities.

In some cases pre-embedding correlative imaging can be quite successful, for example in studies of Jost Enninga (e.g., Mellouk et al, *Cell Host & Microbe* 2014) or Eric Jorgensen (Watanabe et al, *Nature Methods* 2011). Do the authors see a distinction between adherent cell cultures and unsupported tissues or tissue sections?

Other investigators have insisted that FIB-SEM requires especially heavy labelling. What was done differently here to make the light labeling possible? Such clues may be very useful to ongoing developments in the literature. Also, the present protocol skips osmium staining entirely. The authors must have compared images with and without osmium. What visible features do we lose as a result?

3. Significance:

Significance (Required)

Perhaps the greatest challenge to large volume electron microscopy is to deal with rare events. Correlative fluorescence light-electron microscopy effectively addresses the issue of finding the region of interest in a two-dimensional specimen such as a thin section or even a monolayer cell culture. For tissues the solutions are still at large. It is almost always impractical to image an entire organ at the resolution required to see macromolecules (work of Harald Hess being the exception that proves the rule). The issue is especially acute where the imaging is destructive, as in the case of serial block-face and FIB-SEM tomography. MicroCT has been used so far as the method of choice in the work-up to locate the region of interest within a large specimen, but the approach requires expensive equipment and time-consuming analysis. Furthermore, it can provide directional clues solely on the basis of morphology. Fluorescence would be a far simpler tool, and more informative when labeling is directed to specific molecular components. The manuscript of Ronchi et al provides a much-needed demonstration and detailed set of instructions for 3D CLEM en route to FIB-SEM volume imaging. The examples are presented are both convincing and esthetic. Success depended on integration of a number of factors, including changes to the specimen preparation, so the workflow will be very useful. In short, I recommend publication.

Review Commons - Response to Reviewers' Comments

We thank the reviewers for their appreciation of our work and for their constructive feedback. We have addressed their comments in the point-by-point answers below. We provide a largely revised manuscript as well as the plan for new experiments, following requests from the reviewers.

Reviewer #1 (*Evidence, reproducibility and clarity (Required)*):

In their manuscript, Ronchi P. et al. present a thorough and very well detailed workflow for 3D correlative-light and electron microscopy of whole cells in large tissues. Their approach of iterative block trimming and fluorescence imaging combined with laser branding allowed them explore previously inaccessible tissues and questions. They imaged mammary gland organoids, and resolved the organization of the cells in the organoid and mitotic events. They also specifically targeted tracheal terminal cells of a 3rd instar Drosophila larvae labeled with cytoplasmic DsRed to study their ultrastructure, and several Drosophila ovarian follicular cells (FC) where the cytoplasmic motor protein dynein was knocked down (KD) by RNAi. In the tracheal cells, they observed connected secretory vesicles, probably delivering extra-cellular matrix to the trachea tube. They also found that the overall shape of dynein KD FCs is distorted compared to WT, and that the localization of multi-vesicular bodies/endosomes inside the FCs changed from an apical to basal membrane localization. Although the approach is not entirely new, the manuscript certainly paves the way for future studies to obtain ultrastructural information from large specimens and combine it with meaningful fluorescence information, it's also beautiful and polished.

****Minor comments:****

1. The authors state that they (line 145) that they found the optimal concentration of UA and the best compromise between EM contrast and fluorescence preservation. However, no detail is provided as to how these parameters were experimentally determined.

UA concentration can be optimized in a number of ways, including varying incubation temperature and time. We decided to modify the speed at which the temperature was increased after the freeze substitution step at -90°C. We have experimentally compared 3°C/h vs 5°/h (described in the original on-section CLEM protocol by Kukulski et al) and found a considerable difference for some of the samples we used. This is now described in the revision (lines 152-159). While other protocols might work for some samples, we found this protocol to provide good quality imaging with a large variety of samples we have worked with (including some that are not included in the current paper, e.g. gastrulating *Drosophila* embryos or *C. elegans* larvae).

2. More detail as to how the block face was mounted and kept parallel to the glass bottom dish would be helpful.

This is now described in lines 182-185.

Also, what was the optical slice of the confocal and what was the increment in Z?

The information is now included in lines 191-192.

3. Have the authors tried fluorophores with shorter wavelength (like GFP)? And if so, have they estimated the penetration depth in resin? This would be informative because many GFP lines already exist in the Drosophila model.

In the current version, we have limited our study to red fluorescent proteins because UA is autofluorescent in green. This could cause problems when imaging at shorter wavelengths. We have discussed this in lines 442-444.

However, we agree that an analysis of the behavior of GFP in confocal imaging of the block could improve our work and increase the potential applicability of this method. We are therefore planning an experiment to compare the behavior of EGFP and mCherry during confocal imaging of the block. This experiment will be included in a future revised version.

4. In figure 6 i, how did the authors identify the structures to be MBVs close to the basal surface in the mutant seeing as they do not look like the MBVs seen in WT cells?

In both cases, we identified MBVs as vesicles with a clear lumen containing one or more vesicles of homogenous size. We have included a paragraph in the Material & Methods on “multivesicular body quantification” where this is specified (lines 597-599). The only difference between MBVs of WT and KD cells was their size (shown in Fig. 6j,k,l), and therefore the identification was unambiguous.

Similarly, how were the structures identified as endosomes in figure 5f?

We thank the reviewer for pointing this out. We agree that it is impossible to discriminate between endocytic and exocytic vesicles in our static data. We have therefore rephrased this as “membrane trafficking” (line 355, line 358, line 946).

Can the authors quantify the total MBVs in the apical/basal membranes from both RNAi KD and WT?

We have now segmented all MBVs in 5 KD cells and 5 neighboring WT cells in 4 different oocytes. Representative images, as well as a quantitative analysis of the distribution of MBVs, are shown in Fig. 6m-o.

When we segmented MBVs for this analysis, we realized that WT cells showed large MBVs in their apical side (~5-10% of total MBVs) while in KD cells this population was almost completely absent. This is consistent with a role of dynein in MBV fusion. The data are now included in Fig. 6p and described in lines 410-414.

We thank the reviewer for her/his suggestion to have a more rigorous analysis of the MBVs, which allowed us to make another interesting discovery.

5. The motivation/question in the case of Drosophila samples was clear but not so obvious in the case of the mammary gland organoids. It would be nice if the authors could give a bit more information.

We have included a justification for the use of organoids in lines 226-235.

6. In the introduction (line 124). The dimensions are given in microns and millimeters, which can be a bit confusing.

We have changed this (line 132).

7. In the discussion (lines 427-431), "sample preparation protocols compatible with fluorescence preservation have proven satisfactory for FIB-SEM milling and imaging" have also been shown by others (Porrati et al., 2019).

We agree with the reviewer and indeed Porrati et al., 2019 was cited in the introduction. We have not claimed that we have shown this for the first time. For completeness, we cite the paper again in the discussion (line 469).

8. Figure 1:

- It would be helpful if the cell referred to in g was highlighted.

As suggested, we have indicated the cell with an arrowhead.

- Is the cell in (h) the one in g or in a as written?

We apologize for the mistake. It is indeed the one in g. We have corrected this (line 874).

- Is the image in (k) inverted compared to (i)?

The image in k is not inverted compared to i. We are showing raw images of the confocal and FIB-SEM datasets and therefore the two volumes are rotated 90° with respect to each other along the Y axis. As we have realized that this can be confusing for the readers, we have introduced a sentence in Materials and Methods to describe the different orientations between confocal and FIB-SEM datasets (lines 586-589).

Figure 2:

- In panel d it seems that some numbers on the x axis were duplicated.

We apologize for the mistake. We have corrected figure 2.

Figure 5:

- How does the perfect overlap confirm the accuracy of targeting?

We agree with the reviewer that the overlap is not a measure of accuracy. We have removed the sentence from the legend.

- In panel (e) it was not particularly easy to understand what is the basal lamina.

We have manually segmented the 2 basal membranes in different colors. We hope the reviewer will find this representation clearer.

- In panel (g) the fused vesicle is not as clear as the movie. I also found it open to interpretation whether this is in fact a fused vesicle.

We agree with the reviewer that a 3D object can be better appreciated in the stack image sequence rather than in a single 2D image. However, to help the visualization of the event in the figure, we have shown the 3 ortho-slices in a perspective view in Fig. 5g. This was the best representation we have found. The video with the stack will be available to the readers for a better inspection.

We also agree that it is formally impossible to be sure whether the vesicle is in fact releasing material in the apical space or taking it up. Therefore, we describe now the event as “putative site of fusion...” (line 947).

Reviewer #1 (Significance (Required)):

The increasing demand for volume electron microscopy brings a lot of challenges to correlative light and volume electron microscopy workflows. Although the methods used by the authors are not new, their combination is original. The manuscript will certainly contribute to the field of correlative light and volume EM and provide a rather detailed protocol that can be reproduced by others. The workflow is also more efficient than what was previously achieved using x-ray instead of light microscopy (Bushong et al., 2015; Karreman et al., 2016).

We thank the reviewer for the careful examination of our work and for the positive statement. We are aware that many of the methods used have already been described by others, but we believe that their combination is original and very powerful.

Reviewer #2 (Evidence, reproducibility and clarity (Required)):

In this manuscript, Ronchi et al describe a workflow designed to facilitate the identification and downstream relocation of fluorescently tagged regions of interest within millimetre scale samples, ending with focused ion beam SEM acquisition of the target area. The work follows a logical progression, is well thought out, explained, and illustrated, with proof of concept experiments that are followed up by examples of systems where the potential for the application of the workflow in a 'real' biological question is demonstrated.

For me, the title reads better as ...targeting for FIB SEM acquisition...

We have edited the title according to the reviewer's suggestion

I have only minor suggestions for the revision of the manuscript from this initial version. The introduction, and introductory paragraphs for the two model systems would benefit from some revision to make them more concise however.

We have revised and shortened the introduction and introductory paragraphs for the model systems and we hope the reviewer will find it more concise.

****Summary****

Line 22 - omit large.

Done (line 29)

Introduction

Line 66 - It's probably clearer to discuss this concept as conductivity rather than grounding.

We have changed this sentence (line 76)

Line 105 - Peddie and Collinson 2014 is not the correct reference for this statement.

Presumably this is supposed to be Peddie et al 2014?

We thank the reviewer for spotting this mistake. We have changed the citation (line 1110)

Line 124 - The external diameter of the carrier would give 7 mm², but the internal diameter is smaller, so this size is slightly overstated.

We totally agree. The internal diameter of the carrier is 2mm and therefore the area 3.14 mm². We have corrected the statement (line 132).

Results

General comment - I find the use of NxNxN/N nm³ to be a confusing way of expressing the measurements, so would suggest splitting these up to express as: N nm³ or NxNxN nm.

To avoid confusion, we have now opted for: N nm x N nm x N nm.

Line 141 - no water was used in the FS mixture, and so wasn't needed for preservation of fluorescence? Dry/100% acetone? If no water is needed, this detail should be discussed.

We added a clarification of this point (lines 148-150)

Line 142 - could the authors elaborate on the statement about timing and sample types, to give a better understanding of the context.

The sentence referred to other possible applications (e.g. cell monolayers would require shorter FS time). However, as the method described here is aimed at large 3D samples, we

find that longer FS times (72h) are always required. We have therefore removed the sentence (line 151).

Line 150 - on the choice of fluorophores, did the authors examine any shorter wavelengths, or was the decision to use red/far red based on any other evidence? Anecdotally, red and far red fluorophores may offer better preservation and less longevity in this context, but could the authors elaborate on their reasoning behind the choice shown here?

As replied to reviewer 1, point 3:

In the current version, we have limited our study to red fluorescent proteins because UA is autofluorescent in green. This could cause problems when imaging at shorter wavelengths. We have discussed this in lines 442-444.

However, we agree that an analysis of the behavior of GFP in confocal imaging of the block could improve our work and increase the potential applicability of this method. We are therefore planning an experiment to compare the behavior of EGFP and mCherry during confocal imaging of the block. This experiment will be included in a future revised version.

Line 168 - did immersion in water give rise to any distortion of the resin, or is HM20 sufficiently hydrophobic that this was not a concern? Mismatches in refractive indices (resin, water, glass, oil) could also presumably give rise to some small inaccuracies in depth prediction?

We observed a little distortion of the block face, due to hydration during the imaging step. However, as noticed during trimming at the microtome, this distortion was small and we could achieve a flat surface after removing 1-2 μm . Therefore this was not relevant for our measurements. We however now mention this in the discussion (lines 453-455).

Mismatches of refractive indices also introduce inaccuracies, but these aberrations are reduced the closer the target is to the surface. Therefore, our predictions become more precise after each trimming step to approach the target.

Line 169 - was it possible to quantify the increase in signal? If the block is being hydrated, but the block is not absorbing water (re above point), then it must only be surface fluorophores that are hydrated

The quantified increase in fluorescence signal at the surface is now mentioned here (line 187) and can be observed in Fig. 2b. Indeed, only surface fluorophores are hydrated and we argue that this is an important player in the fluorescence intensity increase.

Line 179 - presumably this is a result of the surface of the block being hydrated (re above points). This is mentioned later, but could be explicitly stated here to make the point more strongly.

We now state this also in line 186.

Line 188 - Peddie et al 2014 contains some limited data for mCherry in sections that could be worth mentioning in support of the findings of reduced photobleaching rates

Thank you for pointing this out. We now cite Peddie et al 2014 (line 208-209)

Line 268 - It is not explicitly stated earlier, but multiple targets at similar depths would also be possible, presumably

We have included a sentence to address this possibility (line 292-293)

Discussion

Line 421 - sections cannot be repeatedly imaged without bleaching too much? Please elaborate on this statement to help strengthen the point as it isn't mentioned earlier in the results.

Our experience with in section fluorescence imaging is that fluorescent proteins are not very stable and bleach rather quickly. However, as we have not measured this with the same setup and with the same samples, we do not have a rigorous proof for this statement. As we believe the comparison with sections is not an important point here, we have removed the sentence (line 463)

Line 435 - FIB SEMs and 2Pi systems are not really so 'common' in the sense suggested, so this final statement should be reworded.

We have changed the sentence (lines 475-477)

M&Ms

Line 540 - grooves, not groves

Changed (line 589)

Figure 3 legend

Overall, it's a workflow comprising many methods, so it's best described as a schematic of the workflow.

Changed (line 900)

Confocal panel - target, not targets, and depth is misspelt.

We thank the reviewer for spotting these mistakes. We have corrected the figure

Figure 4 legend

Line 834 - as far as I can see, this is a different organoid that isn't shown in a and b, so this text should be removed.

The organoid is indeed a different one. We meant that the targeting was performed as shown in a and b. However, as the sentence could generate confusion, we have removed it (line 932).

Figure 5 legend

Line 840 - was, not is

Changed (line 937)

Figure 6

a) It would help with clarity to also put e.g. white arrows on the WT epithelium

As we use arrows and arrowheads to indicate different events in the image, we have used green asterisks to label the nucleus of the WT cell and a red asterisk for the KD, as we have done in all the panels in figure 6, where both cell types are present in the same image.

f,g) It isn't really clear on first viewing what these images show, so they would benefit from some labels.

We have added labels to indicate all the cells represented in the images as well as the space in between (VM, vitelline material). Microvilli are now indicated with arrowheads. We have also explained in the figure legend that here we show in detail the structures indicated by black arrows in Fig. 6a, to help give a context to the high mag detail (lines 964-965).

****Minor stylistic comments****

There should be a space between numbers and units; this is inconsistent throughout.

We have corrected this.

The use of black versus white text on the figures is inconsistent.

We have fixed this.

Table 1 - is it in the supplementary material or not? If it is, it should be referenced as such in the text. The formatting could use some refinement to match the standard of the other figures.

The table is supplementary material. We have now referenced it as such and we have reformatted it.

Capitalisation is inconsistent throughout.

We have revised the text.

The manuscript describes a workflow that connects several pre-existing methods to enable precision targeting of individual fluorescently tagged structures within a larger sample volume. The possibility for multi-modal imaging within a single embed specimen facilitates correlation of data for structure, with that of function. The work will be of interest to all scientists in the field of correlative microscopy

We thank the reviewer for her/his positive evaluation.

Reviewer #3 (Evidence, reproducibility and clarity (Required)):

The manuscript is written very clearly overall. I would like to raise a number of issues that the authors might address. Most are at the level of proof-reading.

The workflow still depends on availability of a specialized confocal microscope with two-photon laser excitation for marking the region of interest. A tweak to the method might simply be to scratch or etch markings onto the planed surface near the edges. Provided a motorized stage is available on the light microscope, the region of interest could be located precisely with reference to those, and then relocated in the SEM. It would be enough to suggest this, or another similar method, for those who don't have access to the two-photon microscope.

In our view, the 2pi branding is important to position the FIB-SEM acquisition with high precision, reliability and confidence. However, we agree with the reviewer that there are other approaches to accomplish this task, which we now mention in the text. One is to simply measure the distance from the edges or corners of the block (lines 256-259). Another, could be to manually introduce landmarks (lines 259-260).

The second is to clarify in the text that the top-down view of the confocal microscopy is orthogonal to that of the FIB. This appears as a note in the caption to Figure 1, but it is an important point to align the expectations of readers who are not closely familiar with the methods.

We agree with the reviewer that this is a point that requires further clarification. We have described this in Materials & Methods in the paragraph "Image processing, dataset registration, visualization and segmentation" (lines 586-589).

The legend labels in Figure 1 do not match the figure itself, as if it were recompiled from an earlier draft: g-j) refers next to a).

We apologize for the mistake. We have corrected it.

The decrease in fluorescence intensity with depth into the specimen remains a bit ambiguous. The significant part of the text is dedicated to the suggestion that inherent protein fluorescence is affected by water content in the resin. After cutting back from the surface, are the originally deeper layers still dim, or do they become brighter? In other words, is the effect chemical or optical?

As we wrote in the discussion, probably both optical effects and hydration play a role in the observed fluorescence drop. The hydration we describe probably only takes place on the block surface when dipping the block in water for imaging. Therefore, when we expose deeper layers after removing the resin on top, they do become brighter. However, we cannot completely disentangle the optical and hydration effect. To make this clearer, we have explained the point in more detail in the discussion (lines 452-455). At the same time, we are planning a new experiment to compare the fluorescence signal in the presence or absence of water in the dish, which will allow us to discriminate between the two effects.

Loss of confocal intensity with depth would be expected on the basis of a refractive index mismatch to the design parameters of the objective, especially for high numerical aperture. The objective is specified as multi-immersion but no further details are given. Details of the lense we used are now given in Materials and Methods (lines 539-540)

Another easy test would be to embed fluorescent beads as intensity standards. There could also be absorption of the fluorescence emission by the resin and stain, but such a strong effect in a few tens of microns would suggest that the block is quite dark. That seems inconsistent with the images in supplementary figures. Personally I was not bothered by the dimming in depth, since the conclusions do not depend on quantitative fluorescence intensities.

We agree with the reviewer that, although the fluorescence intensity drop is an effect that is worth describing because it has an implication for the identification of fluorescent targets in the block, our method does not rely on quantitative imaging. In all cases, we were able to detect fluorescence signal even very deep from the block surface and this was enough to target those cells at the FIB-SEM.

In some cases pre-embedding correlative imaging can be quite successful, for example in studies of Jost Enninga (e.g., Mellouk et al, Cell Host & Microbe 2014) or Eric Jorgensen (Watanabe et al, Nature Methods 2011). Do the authors see a distinction between adherent cell cultures and unsupported tissues or tissue sections?

We completely agree with the reviewer that pre-embedding CLEM can be extremely successful and it is a very valuable tool, especially for the study of dynamic event in cell cultures. However, while for adherent cells the targeting is essentially a 2D problem and is facilitated by the fact that cells can be identified on the surface of the block under the SEM beam, for larger 3D samples the situation is much more complicated. We often lack landmarks and surface references and an anisotropic deformation occurs during sample prep, making targeting and localization prediction extremely inaccurate.

Other investigators have insisted that FIB-SEM requires especially heavy labelling. What was done differently here to make the light labeling possible? Such clues may be very useful to ongoing developments in the literature. Also, the present protocol skips osmium staining

entirely. The authors must have compared images with and without osmium. What visible features do we lose as a result?

We provide a detailed freeze substitution protocol in table S1, such that the method can be easily reproduced. Although FIB-SEM imaging of osmium-free samples is not very common, it has been shown by others before (Porrati et al., 2019), with a slightly different FS protocol. We found that our sample preparation is good enough for the detection of all membranous organelles, but also microtubules, centrioles and other subcellular structures. We did not observe any big difference compared to the more standard protocols containing osmium (line 136).

Perhaps the greatest challenge to large volume electron microscopy is to deal with rare events. Correlative fluorescence light-electron microscopy effectively addresses the issue of finding the region of interest in a two-dimensional specimen such as a thin section or even a monolayer cell culture. For tissues the solutions are still at large. It is almost always impractical to image an entire organ at the resolution required to see macromolecules (work of Harald Hess being the exception that proves the rule). The issue is especially acute where the imaging is destructive, as in the case of serial block-face and FIB-SEM tomography. MicroCT has been used so far as the method of choice in the work-up to locate the region of interest within a large specimen, but the approach requires expensive equipment and time-consuming analysis. Furthermore, it can provide directional clues solely on the basis of morphology. Fluorescence would be a far simpler tool, and more informative when labeling is directed to specific molecular components. The manuscript of Ronchi et al provides a much-needed demonstration and detailed set of instructions for 3D CLEM en route to FIB-SEM volume imaging. The examples are presented are both convincing and esthetic. Success depended on integration of a number of factors, including changes to the specimen preparation, so the workflow will be very useful. In short, I recommend publication. We thank the reviewer for the generous comments.

April 27, 2021

RE: JCB Manuscript #202104069T

Dr. Paolo Ronchi
EMBL
Electron Microscopy Core Facility
Meyerhofstrasse 1
Heidelberg 69117
Germany [DE]

Dear Dr. Ronchi,

Thank you for submitting your transfer manuscript entitled "Fluorescence-based 3D targeting of FIB-SEM acquisition of small volumes in large samples". We have discussed your manuscript, rebuttal, and reviews and consulted several members of the JCB Editorial Board for feedback on the manuscript. In our view, the manuscript is highly relevant to the JCB readership since CLEM, FIB-SEM, and related approaches are very important modern imaging techniques. It is a good match for the JCB Tools format, as there is a high need for this type of approach and for easier workflows. We think this manuscript will be of interest to a lot of readers of the journal. The reviews were quite strong and we feel that your plans to address all the concerns are thorough. We will leave it to you to decide whether to include additional experiments as delineated in your response to the Review Commons reviewers.

Pending changes in response to the reviews, we would be happy to publish your paper in JCB pending final revisions necessary to meet our formatting guidelines (see details below).

1) Title: Please consider the following revision suggestions aimed at increasing the accessibility of the work for a broad audience and non-experts.

High-precision region of interest targeting workflow for 3D electron microscopy analyses

2) Figure formatting: Scale bars must be present on all microscopy images, including inset magnifications. Please add scale bars to 5g (magnification)

3) Statistical analysis: Error bars on graphic representations of numerical data must be clearly described in the figure legend. The number of independent data points (n) represented in a graph must be indicated in the legend. Statistical methods should be explained in full in the materials and methods. For figures presenting pooled data the statistical measure should be defined in the figure legends.

4) Materials and methods: Should be comprehensive and not simply reference a previous publication for details on how an experiment was performed. Please provide full descriptions in the text for readers who may not have access to referenced manuscripts.

- For all cell lines, vectors, constructs/cDNAs, etc. - all genetic material: please include database / vendor ID (e.g., Addgene, ATCC, etc.) or if unavailable, please briefly describe their basic genetic features *even if described in other published work or gifted to you by other investigators*
- Please include species and source for all antibodies, including secondary, as well as catalog numbers/vendor identifiers if available.
- Sequences should be provided for all oligos: primers, si/shRNA, gRNAs, etc.
- Microscope image acquisition: The following information must be provided about the acquisition and processing of images:
 - a. Make and model of microscope
 - b. Type, magnification, and numerical aperture of the objective lenses
 - c. Temperature
 - d. imaging medium
 - e. Fluorochromes
 - f. Camera make and model
 - g. Acquisition software
 - h. Any software used for image processing subsequent to data acquisition. Please include details and types of operations involved (e.g., type of deconvolution, 3D reconstitutions, surface or volume rendering, gamma adjustments, etc.).

5) A summary paragraph of all supplemental material should appear at the end of the Materials and methods section.

- Please include one brief sentence per item.

A. MANUSCRIPT ORGANIZATION AND FORMATTING:

Full guidelines are available on our Instructions for Authors page, <https://jcb.rupress.org/submission-guidelines#revised>. **Submission of a paper that does not conform to JCB guidelines will delay the acceptance of your manuscript.**

B. FINAL FILES:

-- High-resolution figure and video files: See our detailed guidelines for preparing your production-ready images, <https://jcb.rupress.org/fig-vid-guidelines>.

**The license to publish form must be signed before your manuscript can be sent to production. A

link to the electronic license to publish form will be sent to the corresponding author only. Please take a moment to check your funder requirements before choosing the appropriate license.**

Thank you for your attention to these final processing requirements. Please revise and format the manuscript and upload materials within 1 month. If complications arising from measures taken to prevent the spread of COVID-19 will prevent you from meeting this deadline (e.g. if you cannot retrieve necessary files from your laboratory, etc.), please let us know and we can work with you to determine a suitable revision period.

Thank you for this interesting contribution, we look forward to publishing your paper in the Journal of Cell Biology.

Sincerely,

Andres Leschziner, PhD
Monitoring Editor, Journal of Cell Biology

Melina Casadio, PhD
Senior Scientific Editor, Journal of Cell Biology

We thank the editors and reviewers for their appreciation of our work and for their constructive feedback. We have addressed their comments in the point-by-point answers below. In the revised manuscript, we include new data that provide a quantitative comparative analysis of the emission of different fluorescent proteins in resin. The new experiments show that red fluorescent proteins are more suited for our approach, although mEGFP can also be detected at least close to the block surface. This analysis is a reference for the applicability of the method and it helps to clarify some discussion points in response to the reviewers' comments.

The manuscript has been largely revised and it matches now JCB formatting guidelines.

Response to the editor:

1) Title: Please consider the following revision suggestions aimed at increasing the accessibility of the work for a broad audience and non-experts.

High-precision region of interest targeting workflow for 3D electron microscopy analyses
We have modified the title to: „High-precision targeting workflow for volume electron microscopy“.

2) Figure formatting: Scale bars must be present on all microscopy images, including inset magnifications. Please add scale bars to 5g (magnification)

We thank the editor for pointing it out. We have included a scale bar in the inset of fig. 5G.

3) Statistical analysis: Error bars on graphic representations of numerical data must be clearly described in the figure legend. The number of independent data points (n) represented in a graph must be indicated in the legend. Statistical methods should be explained in full in the materials and methods. For figures presenting pooled data the statistical measure should be defined in the figure legends.

We have revised the text and the information regarding all the statistical analyses is included.

4) Materials and methods: Should be comprehensive and not simply reference a previous publication for details on how an experiment was performed. Please provide full descriptions in the text for readers who may not have access to referenced manuscripts.

*- For all cell lines, vectors, constructs/cDNAs, etc. - all genetic material: please include database / vendor ID (e.g., Addgene, ATCC, etc.) or if unavailable, please briefly describe their basic genetic features *even if described in other published work or gifted to you by other investigators**

- Please include species and source for all antibodies, including secondary, as well as catalog numbers/vendor identifiers if available.

- Sequences should be provided for all oligos: primers, si/shRNA, gRNAs, etc.

- Microscope image acquisition: The following information must be provided about the acquisition and processing of images:

a. Make and model of microscope

b. Type, magnification, and numerical aperture of the objective lenses

c. Temperature

d. imaging medium

e. Fluorochromes

f. Camera make and model

g. Acquisition software

h. Any software used for image processing subsequent to data acquisition. Please include details and types of operations involved (e.g., type of deconvolution, 3D reconstitutions,

surface or volume rendering, gamma adjustments, etc.).

We believe our current version contains all the required information.

5) A summary paragraph of all supplemental material should appear at the end of the Materials and methods section.

- Please include one brief sentence per item.

We have included the paragraph.

Response to reviewers:

Reviewer #1 (Evidence, reproducibility and clarity (Required)):

In their manuscript, Ronchi P. et al. present a thorough and very well detailed workflow for 3D correlative-light and electron microscopy of whole cells in large tissues. Their approach of iterative block trimming and fluorescence imaging combined with laser branding allowed them explore previously inaccessible tissues and questions. They imaged mammary gland organoids, and resolved the organization of the cells in the organoid and mitotic events. They also specifically targeted tracheal terminal cells of a 3rd instar Drosophila larvae labeled with cytoplasmic DsRed to study their ultrastructure, and several Drosophila ovarian follicular cells (FC) where the cytoplasmic motor protein dynein was knocked down (KD) by RNAi. In the tracheal cells, they observed connected secretory vesicles, probably delivering extra-cellular matrix to the trachea tube. They also found that the overall shape of dynein KD FCs is distorted compared to WT, and that the localization of multi-vesicular bodies/endosomes inside the FCs changed from an apical to basal membrane localization. Although the approach is not entirely new, the manuscript certainly paves the way for future studies to obtain ultrastructural information from large specimens and combine it with meaningful fluorescence information, it's also beautiful and polished.

****Minor comments:****

1. The authors state that they (line 145) that they found the optimal concentration of UA and the best compromise between EM contrast and fluorescence preservation. However, no detail is provided as to how these parameters were experimentally determined.

UA concentration can be optimized in a number of ways, including varying incubation temperature and time. We decided to modify the speed at which the temperature was increased after the freeze substitution step at -90°C. We have experimentally compared 3°C/h vs 5°/h (described in the original on-section CLEM protocol by Kukulski et al) and found a considerable difference for some of the samples we used. This is now described in the revision (lines 140-148). While other protocols might work for some samples, we found this protocol to provide good quality imaging with a large variety of samples we have worked with (including some that are not included in the current paper, e.g. gastrulating *Drosophila* embryos or *C. elegans* larvae).

2. More detail as to how the block face was mounted and kept parallel to the glass bottom dish would be helpful.

A more detailed description of how the block is prepared and mounted for confocal imaging is now included in lines 202-210.

Also, what was the optical slice of the confocal and what was the increment in Z?

The information is now included in lines 213-215.

3. Have the authors tried fluorophores with shorter wavelength (like GFP)? And if so, have they estimated the penetration depth in resin? This would be informative because many GFP lines already exist in the Drosophila model.

In this revised version we have included a full characterization of the behavior of mCherry and EGFP in block. To this aim, we have prepared a mixed suspension of HeLa cells expressing H2B-mCherry or H2B-EGFP and followed our FPF/FS protocol. We prepared the block as described in the paper and imaged it at the confocal. In this set of experiments, we have measured: i) the fluorescence loss due to sample preparation, by comparing the laser intensities needed to image embedded and non-embedded samples, ii) the contribution of hydration in fluorescence intensity and iii) the dependence of the intensity from distance from the surface, for both fluorescent proteins (FPs). For these experiments, we used a high cell density so that with each confocal slice we were able to image several nuclei expressing mCherry or EGFP and therefore allowing a direct quantitative comparison of the behavior of the two FPs. As the data in figure 2 show, mCherry is much more suitable for our method, even though bright EGFP signals can be detected, especially close to the block surface and after long hydration time.

The results are shown in figure 2 and described in lines 162-192.

4. In figure 6 i, how did the authors identify the structures to be MBVs close to the basal surface in the mutant seeing as they do not look like the MBVs seen in WT cells?

In both cases, we identified MVBs as vesicles with a clear lumen containing one or more vesicles of homogenous size. We have included a paragraph in the Material & Methods on “multivesicular body quantification” where this is specified (lines 590-595). The only difference between MVBs of WT and KD cells was their size (shown in Fig. 6J,K,L), and therefore the identification was unambiguous.

Similarly, how were the structures identified as endosomes in figure 5f?

We thank the reviewer for pointing this out. We agree that it is impossible to discriminate between endocytic and exocytic vesicles in our static data. We have therefore rephrased this as “membrane trafficking” (lines 320-326, lines 965-967).

Can the authors quantify the total MVBs in the apical/basal membranes from both RNAi KD and WT?

We have now segmented all MVBs in 5 KD cells and 5 neighboring WT cells in 4 different oocytes. Representative images, as well as a quantitative analysis of the distribution of MVBs, are shown in Fig. 6M-O.

When we segmented MVBs for this analysis, we realized that WT cells showed large MVBs in their apical side (~5-10% of total MVBs) while in KD cells this population was almost completely absent. This is consistent with a role of dynein in MVB fusion. The data are now included in Fig. 6P and described in lines 379-383.

We thank the reviewer for her/his suggestion to have a more rigorous analysis of the MVBs, which allowed us to make another interesting discovery.

5. The motivation/question in the case of Drosophila samples was clear but not so obvious in the case of the mammary gland organoids. It would be nice if the authors could give a bit more information.

The organoids were used a challenging model to set up our strategies. Even though targeting organoids or single cells within an organoid can be of primary interest for many biological

studies, in our case the reason why we chose this model was essentially a technical one. This is now described in lines 196-201.

6. *In the introduction (line 124). The dimensions are given in microns and millimeters, which can be a bit confusing.*

We have changed this (line 119).

7. *In the discussion (lines 427-431), "sample preparation protocols compatible with fluorescence preservation have proven satisfactory for FIB-SEM milling and imaging" have also been shown by others (Porrati et al., 2019).*

We agree with the reviewer and indeed Porrati et al., 2019 was cited in the introduction. We have not claimed that we have shown this for the first time. For completeness, we cite the paper again in the discussion (line 428).

8. *Figure 1:*

- *It would be helpful if the cell referred to in g was highlighted.*

As suggested, we have indicated the cell with an arrowhead.

- *Is the cell in (h) the one in g or in a as written?*

We apologize for the mistake. We have corrected this (line 891-892). As we have introduced other panels to this figure, the relevant panels are now L (FIBSEM) and K (fluorescence).

- *Is the image in (k) inverted compared to (i)?*

The image in the FIB-SEM view (now P) is not inverted compared to LM (O). We are showing raw images of the confocal and FIB-SEM datasets and therefore the two volumes are rotated 90 degrees with respect to each other along the Y axis. This applies to all the panels of figure 1. As we have realized that this can be confusing for the readers, we have introduced a sentence in Materials and Methods to describe the different orientations between confocal and FIB-SEM datasets (lines 580-583).

Figure 2:

- *In panel d it seems that some numbers on the x axis were duplicated.*

We apologize for the mistake. We have corrected the figure, which is now Fig. S1.

Figure 5:

- *How does the perfect overlap confirm the accuracy of targeting?*

We agree with the reviewer that the overlap is not a measure of accuracy. We have removed the sentence from the legend.

- *In panel (e) it was not particularly easy to understand what is the basal lamina.*

We have manually segmented the 2 basal membranes in different colors. We hope the reviewer will find this representation clearer.

- *In panel (g) the fused vesicle is not as clear as the movie. I also found it open to interpretation whether this is in fact a fused vesicle.*

We agree with the reviewer that a 3D object can be better appreciated in the stack image sequence rather than in a single 2D image. However, to help the visualization of the event in the figure, we have shown the 3 ortho-slices in a perspective view in Fig. 5G. This was the

best representation we have found. The video with the stack will be available to the readers for a better inspection.

We also agree that it is formally impossible to be sure whether the vesicle is in fact releasing material in the apical space or taking it up. Therefore, we describe now the event as “putative site of fusion...” (line 967-969).

Reviewer #1 (Significance (Required)):

The increasing demand for volume electron microscopy brings a lot of challenges to correlative light and volume electron microscopy workflows. Although the methods used by the authors are not new, their combination is original. The manuscript will certainly contribute to the field of correlative light and volume EM and provide a rather detailed protocol that can be reproduced by others. The workflow is also more efficient than what was previously achieved using x-ray instead of light microscopy (Bushong et al., 2015; Karreman et al., 2016).

We thank the reviewer for the careful examination of our work and for the positive statement. We are aware that many of the methods used have already been described by others, but we believe that their combination is original and very powerful.

Reviewer #2 (Evidence, reproducibility and clarity (Required)):

In this manuscript, Ronchi et al describe a workflow designed to facilitate the identification and downstream relocation of fluorescently tagged regions of interest within millimetre scale samples, ending with focused ion beam SEM acquisition of the target area. The work follows a logical progression, is well thought out, explained, and illustrated, with proof of concept experiments that are followed up by examples of systems where the potential for the application of the workflow in a 'real' biological question is demonstrated.

For me, the title reads better as ...targeting for FIB SEM acquisition...

We have edited the title according to the reviewer's suggestion

I have only minor suggestions for the revision of the manuscript from this initial version. The introduction, and introductory paragraphs for the two model systems would benefit from some revision to make them more concise however.

We have revised and shortened the introduction and introductory paragraphs for the model systems and we hope the reviewer will find it more concise.

****Summary****

Line 22 - omit large.

Done (line 27)

Introduction

Line 66 - It's probably clearer to discuss this concept as conductivity rather than grounding.

We have changed this sentence (line 70)

Line 105 - Peddie and Collinson 2014 is not the correct reference for this statement.

Presumably this is supposed to be Peddie et al 2014?

We thank the reviewer for spotting this mistake. We have changed the citation (line 101)

Line 124 - The external diameter of the carrier would give 7 mm², but the internal diameter is smaller, so this size is slightly overstated.

We totally agree. The internal diameter of the carrier is 2mm and therefore the area 3.14 mm². We have corrected the statement (line 119).

Results

General comment - I find the use of NxNxN/N nm³ to be a confusing way of expressing the measurements, so would suggest splitting these up to express as: N nm³ or NxNxN nm.

To avoid confusion, we have now opted for: N nm x N nm x N nm.

Line 141 - no water was used in the FS mixture, and so wasn't needed for preservation of fluorescence? Dry/100% acetone? If no water is needed, this detail should be discussed.

We added a clarification of this point (lines 136-139)

Line 142 - could the authors elaborate on the statement about timing and sample types, to give a better understanding of the context.

The sentence referred to other possible applications (e.g. cell monolayers would require shorter FS time). However, as the method described here is aimed at large 3D samples, we find that longer FS times (72h) are always required. We have therefore removed the sentence.

Line 150 - on the choice of fluorophores, did the authors examine any shorter wavelengths, or was the decision to use red/far red based on any other evidence? Anecdotally, red and far red fluorophores may offer better preservation and less longevity in this context, but could the authors elaborate on their reasoning behind the choice shown here?

As replied to reviewer 1, point 3:

In this revised version we have included a full characterization of the behavior of mCherry and EGFP in block. To this aim, we have prepared a mixed suspension of HeLa cells expressing H2B-mCherry or H2B-EGFP and followed our FPF/FS protocol. We prepared the block as described in the paper and imaged it at the confocal. In this set of experiments, we have measured: i) the fluorescence loss due to sample preparation, by comparing the laser intensities needed to image embedded and non-embedded samples, ii) the contribution of hydration in fluorescence intensity and iii) the dependence of the intensity from distance from the surface, for both fluorescent proteins (FPs). For these experiments, we used a high cell density so that with each confocal slice we were able to image several nuclei expressing mCherry or EGFP and therefore allowing a direct quantitative comparison of the behavior of the two FPs. As the data in figure 2 show, mCherry is much more suitable for our method, even though bright EGFP signals can be detected, especially close to the block surface and after long hydration time.

The results are shown in figure 2 and described in lines 162-192.

Line 168 - did immersion in water give rise to any distortion of the resin, or is HM20 sufficiently hydrophobic that this was not a concern? Mismatches in refractive indices (resin, water, glass, oil) could also presumably give rise to some small inaccuracies in depth prediction?

We observed a little distortion of the block face, due to hydration during the imaging step. However, as noticed during trimming at the microtome, this distortion was small and we

could achieve a flat surface after removing 1-2 μm . Therefore this was not relevant for our measurements. We however now mention this in the discussion (lines 419-421). Mismatches of refractive indices also introduce inaccuracies, but these aberrations are reduced the closer the target is to the surface. Therefore, our predictions become more precise after each trimming step to approach the target.

Line 169 - was it possible to quantify the increase in signal? If the block is being hydrated, but the block is not absorbing water (re above point), then it must only be surface fluorophores that are hydrated

The effect of hydration on the fluorescence is now measured for both EGFP and mCherry in figure 2.

Line 179 - presumably this is a result of the surface of the block being hydrated (re above points). This is mentioned later, but could be explicitly stated here to make the point more strongly.

We thank the reviewer for the suggestion. This part of the manuscript has now changed significantly and the effect of hydration on the FPs has been better described.

Line 188 - Peddie et al 2014 contains some limited data for mCherry in sections that could be worth mentioning in support of the findings of reduced photobleaching rates

Thank you for pointing this out. We now cite Peddie et al 2014 (line 182)

Line 268 - It is not explicitly stated earlier, but multiple targets at similar depths would also be possible, presumably

We have included a sentence to address this possibility (line 266-267)

Discussion

Line 421 - sections cannot be repeatedly imaged without bleaching too much? Please elaborate on this statement to help strengthen the point as it isn't mentioned earlier in the results.

Our experience with in section fluorescence imaging is that fluorescent proteins are not very stable and bleach rather quickly. However, as we have not measured this with the same setup and with the same samples, we do not have a rigorous proof for this statement. As we believe the comparison with sections is not an important point here, we have removed the sentence.

Line 435 - FIB SEMs and 2Pi systems are not really so 'common' in the sense suggested, so this final statement should be reworded.

We have changed the sentence (lines 433-435)

M&Ms

Line 540 - grooves, not groves

Changed (line 583)

Figure 3 legend

Overall, it's a workflow comprising many methods, so it's best described as a schematic of the workflow.

Changed (line 925)

Confocal panel - target, not targets, and depth is misspelt.

We thank the reviewer for spotting these mistakes. We have corrected the figure

Figure 4 legend

Line 834 - as far as I can see, this is a different organoid that isn't shown in a and b, so this text should be removed.

The organoid is indeed a different one. We meant that the targeting was performed as shown in a and b. However, as the sentence could generate confusion, we have removed it.

Figure 5 legend

Line 840 - was, not is

The text has been edited here.

Figure 6

a) It would help with clarity to also put e.g. white arrows on the WT epithelium

As we use arrows and arrowheads to indicate different events in the image, we have used green asterisks to label the nucleus of the WT cell and a red asterisk for the KD, as we have done in all the panels in figure 6, where both cell types are present in the same image.

f,g) It isn't really clear on first viewing what these images show, so they would benefit from some labels.

We have added labels to indicate all the cells represented in the images as well as the space in between (VM, vitelline material). Microvilli are now indicated with arrowheads. We have also explained in the figure legend that here we show in detail the structures indicated by black arrows in Fig. 6A, to help give a context to the high mag detail (line 980-983).

Minor stylistic comments

There should be a space between numbers and units; this is inconsistent throughout.

We have corrected this.

The use of black versus white text on the figures is inconsistent.

We have fixed this.

Table 1 - is it in the supplementary material or not? If it is, it should be referenced as such in the text. The formatting could use some refinement to match the standard of the other figures.

The table is supplementary material. We have now referenced it as such and we have reformatted it.

Capitalisation is inconsistent throughout.

We have revised the text.

The manuscript describes a workflow that connects several pre-existing methods to enable precision targeting of individual fluorescently tagged structures within a larger sample volume. The possibility for multi-modal imaging within a single embed specimen facilitates correlation of data for structure, with that of function. The work will be of interest to all scientists in the field of correlative microscopy

We thank the reviewer for her/his positive evaluation.

Reviewer #3 (Evidence, reproducibility and clarity (Required)):

The manuscript is written very clearly overall. I would like to raise a number of issues that the authors might address. Most are at the level of proof-reading.

The workflow still depends on availability of a specialized confocal microscope with two-photon laser excitation for marking the region of interest. A tweak to the method might simply be to scratch or etch markings onto the planed surface near the edges. Provided a motorized stage is available on the light microscope, the region of interest could be located precisely with reference to those, and then relocated in the SEM. It would be enough to suggest this, or another similar method, for those who don't have access to the two-photon microscope. In our view, the 2pi branding is important to position the FIB-SEM acquisition with high precision, reliability and confidence. However, we agree with the reviewer that there are other approaches to accomplish this task, which we now mention in the text. One is to simply measure the distance from the edges or corners of the block (lines 234-236). Another, could be to manually introduce landmarks (lines 236-237).

The second is to clarify in the text that the top-down view of the confocal microscopy is orthogonal to that of the FIB. This appears as a note in the caption to Figure 1, but it is an important point to align the expectations of readers who are not closely familiar with the methods.

We agree with the reviewer that this is a point that requires further clarification. We have described this in Materials & Methods in the paragraph "Image processing, dataset registration, visualization and segmentation" (lines 580-583).

The legend labels in Figure 1 do not match the figure itself, as if it were recompiled from an earlier draft: g-j) refers next to a).

We apologize for the mistake. We have corrected it.

The decrease in fluorescence intensity with depth into the specimen remains a bit ambiguous. The significant part of the text is dedicated to the suggestion that inherent protein fluorescence is affected by water content in the resin. After cutting back from the surface, are the originally deeper layers still dim, or do they become brighter? In other words, is the effect chemical or optical?

With the new experiments provided in this revised version, we have data to prove that hydration is key to increase the emission of fluorescent proteins. Light scattering in the resin contributes to some extent to the fluorescence intensity drop inside the block, but figure 2B shows that water alone can increase the intensity up to 7 times.

Loss of confocal intensity with depth would be expected on the basis of a refractive index mismatch to the design parameters of the objective, especially for high numerical aperture. The objective is specified as multi-immersion but no further details are given.

Details of the lense we used are now given in Materials and Methods (lines 509-510)

Another easy test would be to embed fluorescent beads as intensity standards. There could also be absorption of the fluorescence emission by the resin and stain, but such a strong effect in a few tens of microns would suggest that the block is quite dark. That seems inconsistent with the images in supplementary figures. Personally I was not bothered by the dimming in depth, since the conclusions do not depend on quantitative fluorescence intensities.

We agree with the reviewer that, although the fluorescence intensity drop is an effect that is worth describing because it has an implication for the identification of fluorescent targets in the block, our method does not rely on quantitative imaging. In all cases, we were able to

detect red fluorescence signal even very deep from the block surface and this was enough to target those cells at the FIB-SEM.

In some cases pre-embedding correlative imaging can be quite successful, for example in studies of Jost Enninga (e.g., Mellouk et al, Cell Host & Microbe 2014) or Eric Jorgensen (Watanabe et al, Nature Methods 2011). Do the authors see a distinction between adherent cell cultures and unsupported tissues or tissue sections?

We completely agree with the reviewer that pre-embedding CLEM can be extremely successful and it is a very valuable tool, especially for the study of dynamic event in cell cultures. However, while for adherent cells the targeting is essentially a 2D problem and is facilitated by the fact that cells can be identified on the surface of the block under the SEM beam, for larger 3D samples the situation is much more complicated. We often lack landmarks and surface references and an anisotropic deformation occurs during sample prep, making targeting and localization prediction extremely inaccurate.

Other investigators have insisted that FIB-SEM requires especially heavy labelling. What was done differently here to make the light labeling possible? Such clues may be very useful to ongoing developments in the literature. Also, the present protocol skips osmium staining entirely. The authors must have compared images with and without osmium. What visible features do we lose as a result?

We provide a detailed freeze substitution protocol in table S1, such that the method can be easily reproduced. Although FIB-SEM imaging of osmium-free samples is not very common, it has been shown by others before (Porrati et al., 2019), with a slightly different FS protocol. We found that our sample preparation is good enough for the detection of all membranous organelles, but also microtubules, centrioles and other subcellular structures. We did not observe any big difference compared to the more standard protocols containing osmium (lines 121-125).

Perhaps the greatest challenge to large volume electron microscopy is to deal with rare events. Correlative fluorescence light-electron microscopy effectively addresses the issue of finding the region of interest in a two-dimensional specimen such as a thin section or even a monolayer cell culture. For tissues the solutions are still at large. It is almost always impractical to image an entire organ at the resolution required to see macromolecules (work of Harald Hess being the exception that proves the rule). The issue is especially acute where the imaging is destructive, as in the case of serial block-face and FIB-SEM tomography. MicroCT has been used so far as the method of choice in the work-up to locate the region of interest within a large specimen, but the approach requires expensive equipment and time-consuming analysis. Furthermore, it can provide directional clues solely on the basis of morphology. Fluorescence would be a far simpler tool, and more informative when labeling is directed to specific molecular components. The manuscript of Ronchi et al provides a much-needed demonstration and detailed set of instructions for 3D CLEM en route to FIB-SEM volume imaging. The examples are presented are both convincing and esthetic. Success depended on integration of a number of factors, including changes to the specimen preparation, so the workflow will be very useful. In short, I recommend publication. We thank the reviewer for the generous comments.